eLife RESEARCH ARTICLE

# Rab11 suppresses neuronal stress signaling by localizing dual leucine zipper kinase to axon terminals for protein turnover

Seung Mi Kim, Yaw Quagraine, Monika Singh, Jung Hwan Kim*

Department of Biology, University of Nevada Reno, Reno, United States

## eLife assessment

This **important** manuscript shows that axonal transport of Wnd is required for its normal degradation by the Hiw ubiquitin ligase pathway. In Hiw mutants, the Wnd protein accumulates in nerve terminals. In the absence of axonal transport, Wnd levels also rise and lead to excessive JNK signaling, disrupting neuronal function. These are interesting findings supported by **convincing** data. However, how Rab11 is involved in Golgi processing or axonal transport of Wnd is not resolved as it is clear that Rab11 is not travelling with Wnd to the axon.

*For correspondence:
jungkim@unr.edu

**Competing interest:** The authors declare that no competing interests exist.

**Abstract** Dual leucine zipper kinase (DLK) mediates multiple neuronal stress responses, and its expression levels are constantly suppressed to prevent excessive stress signaling. We found that Wallenda (Wnd), the *Drosophila* ortholog of DLK, is highly enriched in the axon terminals of *Drosophila* sensory neurons in vivo and that this subcellular localization is necessary for Highwire-mediated Wnd protein turnover under normal conditions. Our structure-function analysis found that Wnd palmitoylation is essential for its axon terminal localization. Palmitoylation-defective Wnd accumulated in neuronal cell bodies, exhibited dramatically increased protein expression levels, and triggered excessive neuronal stress responses. Defective intracellular transport is implicated in neurodegenerative conditions. Comprehensive dominant-negative Rab protein screening identified Rab11 as an essential factor for Wnd localization in axon terminals. Consequently, *Rab11* loss-of-function increased the protein levels of Wnd and induced neuronal stress responses. Inhibiting Wnd activity significantly ameliorated neuronal loss and c-Jun N-terminal kinase signaling triggered by *Rab11* loss-of-function. Taken together, these suggest that DLK proteins are constantly transported to axon terminals for protein turnover and a failure of such transport can lead to neuronal loss. Our study demonstrates how subcellular protein localization is coupled to protein turnover for neuronal stress signaling.

## Introduction

A typical neuron is composed of distinct somatodendritic and axonal compartments that express unique compositions of membrane and cytoplasmic proteins, which are precisely established and maintained through a complex network of intracellular transport processes (*Guedes-Dias and Holzbaur, 2019*; *Guillaud et al., 2020*). Consistently, impaired neuronal intracellular transport has been recognized as an early cellular event in various neurodegenerative disorders including amyotrophic lateral sclerosis (*Collard et al., 1995*; *De Vos and Hafezparast, 2017*), Alzheimer's (*Calkins et al., 2011*; *Wang et al., 2015*), Parkinson's (*Saha et al., 2004*; *Chu et al., 2012*), and Huntington's diseases

(*Lee et al., 2004*; *Reddy and Shirendeb, 2012*; *Smith et al., 2014*). The intricate regulation of intracellular transport relies on a family of small G proteins known as Rab (Ras-related in the brain) proteins (*Stenmark, 2009*). Studies have demonstrated altered Rab activity or expression levels in Huntington's disease model (*Li et al., 2009a*; *Li et al., 2009b*), Alzheimer's disease model (*Pensalfini et al., 2020*), and from the amyotrophic lateral sclerosis patients (*Mitra et al., 2019*). However, how defective neuronal intracellular transport leads to neuronal loss is not fully understood.

DLK, also known as MAP3K12 is a cytoplasmic protein kinase essential for stress signaling and mediates diverse neuronal responses including axon regeneration, axon degeneration, and cell death (*Miller et al., 2009*; *Xiong et al., 2010*; *Shin et al., 2012*; *Xiong and Collins, 2012*; *Watkins et al., 2013*; *Li et al., 2021*). Under normal conditions, neurons tightly control DLK activity to prevent excessive stress signaling primarily through protein turnover by an evolutionarily conserved PHR1/RPM-1/Highwire (Hiw) ubiquitin ligases (*Nakata et al., 2005*; *Collins et al., 2006*; *Xiong et al., 2010*; *Huntwork-Rodriguez et al., 2013*). Notably, recent studies have shown potential widespread involvement of DLK in neuronal death observed in neurotrophic factor deprivation, glaucoma, amyotrophic lateral sclerosis, and Alzheimer's disease models (*Pozniak et al., 2013*; *Welsbie et al., 2013*; *Welsbie et al., 2017*; *Larhammar et al., 2017a*; *Le Pichon et al., 2017*; *Patel et al., 2017*). These observations suggest the possibility of dysregulated or elevated DLK protein/activity levels in these conditions. Interestingly, DLK proteins are found within axonal compartments (*Xiong et al., 2010*; *Klinedinst et al., 2013*; *Holland et al., 2016*; *Niu et al., 2020*; *Niu et al., 2022*). However, there is limited understanding of the mechanisms underlying DLK protein localization in axonal compartments and how such axonal localization of DLK plays a role in neuronal stress responses.

Here, we report a novel coupling between axon terminal localization and protein turnover of DLK. We further found that Rab11 is essential for this coupling, which when disrupted, leads to neuronal loss by excessive stress signaling. We found that Wallenda (Wnd), *Drosophila* DLK is highly enriched in the axon terminals of *Drosophila* sensory neurons in vivo. Furthermore, we found that Wnd protein turnover by Hiw is confined to axon terminals. We identified a protein palmitoylation on Wnd as an essential motif for Wnd protein localization in axon terminals. By taking advantage of this unique behavior of *Drosophila* DLK, Wnd in its reliance on protein palmitoylation for axonal localization, we demonstrated that Wnd protein turnover is greatly diminished when Wnd proteins are not transported to axon terminals, which in turn causes excessive Wnd signaling and neuronal cell death. We further uncovered the selective role of Rab11 in Wnd axonal localization and protein turnover. Importantly, neuronal stress responses caused by *Rab11* loss-of-function were significantly mitigated by inhibiting Wnd. Together, our findings suggest that Rab11 suppresses DLK protein levels by coupling axonal localization and protein turnover of DLK.

## Results

### Wnd is highly enriched in axon terminals and Wnd protein turnover is restricted to axon terminals

DLK proteins are found in the axonal compartments of diverse animals (*Holland et al., 2016*; *Niu et al., 2020*; *Niu et al., 2022*). To understand the mechanism of DLK axonal localization, we first set out to investigate the extent of axonal localization of *Drosophila* DLK, Wnd. We took advantage of a *wnd* protein-trap allele, *Mi{MIC}wnd^GFSTF^*, which allows us to visualize endogenous Wnd protein localization via an EGFP tag (*Venken et al., 2011*; *Li et al., 2017*). The levels of Wnd-GFSTF proteins in the third instar larva stage were extremely low (*Figure 1A*, left). To enhance the detection of Wnd-GFSTF, the experiment was performed in *hiw* mutants, *hiw^ΔN^* (*Wu et al., 2005*). Hiw is a neuronal ubiquitin ligase that selectively targets Wnd for protein degradation (*Collins et al., 2006*). Immunostaining was conducted using an anti-GFP antibody, and the larval samples were examined using a confocal microscope. The result showed a strikingly polarized expression pattern of Wnd. While Wnd-GFSTF expression was evident in the neuropil region in the ventral nerve cord (VNC) (*Figure 1A and B*), we could not detect any meaningful signal from neuronal cell bodies, especially from the peripheral nervous system neurons. Neuropils are enriched in neuronal dendrites and axons, while neuronal cell bodies are mostly absent in this region. This suggests that Wnd protein expression levels are kept extremely low in neuronal cell bodies but enriched in neuronal processes likely in axon terminals.

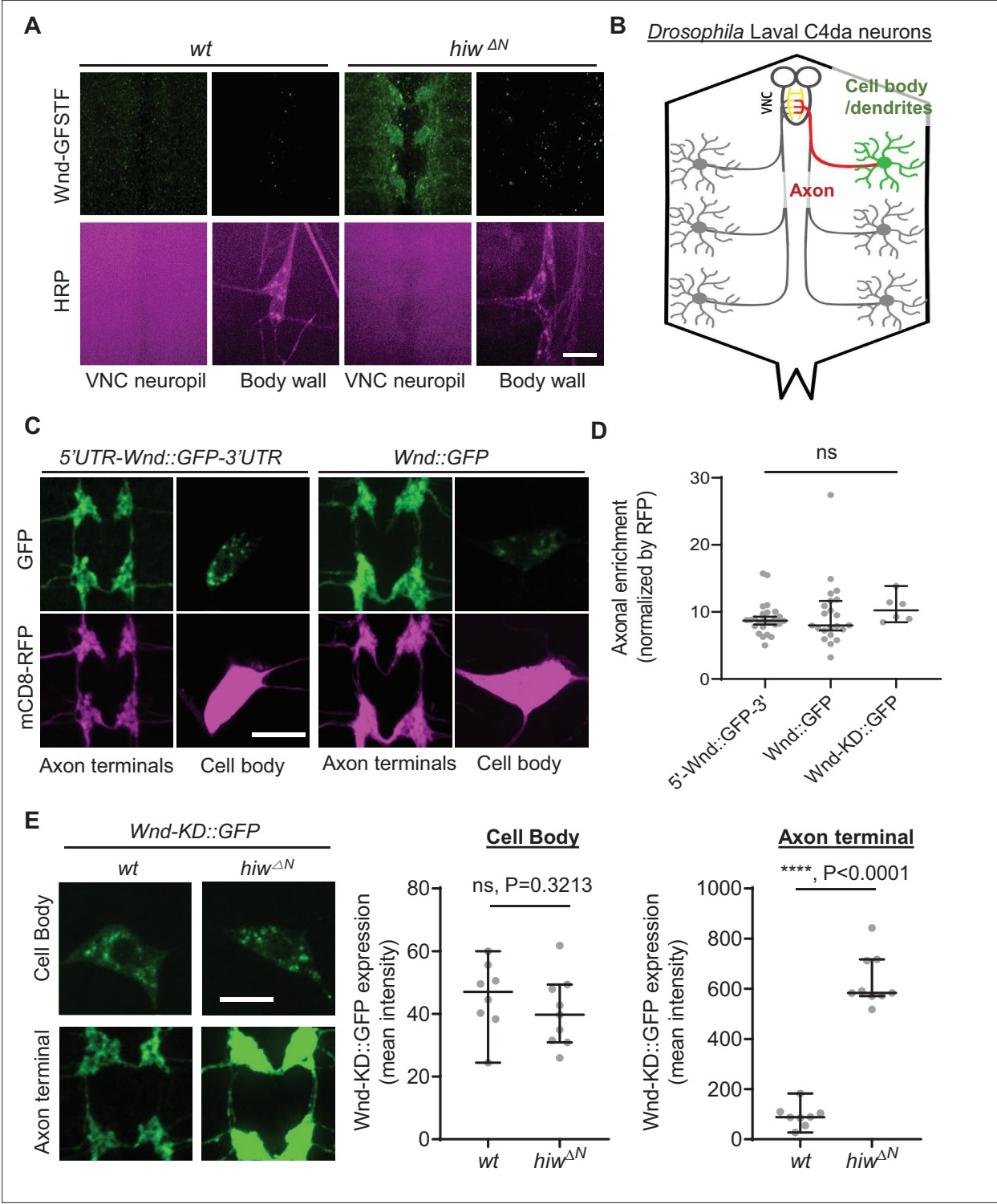

**Figure 1.** Wallenda (Wnd) protein turnover by Highwire (Hiw) is restricted in the axon terminals of *Drosophila* sensory neurons. Wnd is highly enriched in axon terminals (**A–D**). (**A**) A representative image of *Mi{MIC}wnd* ^GFSTF^ in the neuropil of larval ventral nerve cords (top) and C4da neuron cell body (bottom) from wild-type male (*w^1118^*) or hemizygous male *hiw* mutant (*hiw^ΔN^*) were shown. Wnd-GFP (Wnd-GFSTF, green) and anti-HRP staining (magenta). Scale bar = 10 µm. (**B**) Schematic of *Drosophila* larval C4da neurons. C4da cell bodies and dendrites are located in the body wall (green) while their axons (red) terminate in the ventral nerve cord (VNC) collectively forming the ladder-like structure of axon terminals (yellow). (**C**) The Wnd::GFP (with or without *wnd*-UTRs, green) along with mCD8::RFP (magenta) were expressed in the larval C4da neurons in the presence of a dual leucine zipper kinase (DLK) inhibitor, DLKi. The expression levels of transgenes were shown in C4da cell body and the axon terminals. (**D**) The axonal enrichment index was expressed as median ± 95% CI. Sample numbers were 5'-Wnd::GFP-3' (*w^1118^; UAS-wnd-5'UTR-Wnd::GFP-wnd-3'UTR/+;ppk-GAL4,*

*Figure 1 continued on next page*

*Figure 1 continued*

*UAS-mCD8::RFP/+*) (n=26), Wnd::GFP (*w[1118]; UAS-Wnd::GFP-SV40 3'UTR /+; ppk-GAL4, UAS-mCD8::RFP/+*) (n=22), Wnd-KD::GFP (*w[1118]; UAS-Wnd-KD::GFP/+; ppk-GAL4, UAS-mCD8::RFP/+*) (n=6). All samples were treated with DLKi; One-way ANOVA (F(2,51) = 0.5176) followed by post hoc Tukey's multiple comparison test. (**E**) The Wnd-KD::GFP transgene along with mCD8::RFP was expressed in C4da neurons with the *ppk-Gal4* driver, in wild-type (*wt*) (*w[1118];UAS-Wnd-KD::GFP/+;ppk-Gal4, UAS-mCD8::RFP/+,* sample n=8) or in *hiw* mutant (*hiw[ΔN]; UAS-Wnd-KD::GFP/+;ppk-Gal4,UAS-mCD8::RFP/+,* sample n=9) larvae. Representative images of Wnd-KD::GFP in C4da cell bodies and axon terminals, stained for anti-GFP (green). The average Wnd-KD::GFP intensity from cell bodies and axon terminals were quantified and expressed as median ± 95% CI; (Cell body: U=25, p=0.3213, Axon terminal: U=0, p<0.0001, two-tailed Mann-Whitney test). Scale bar = 10 μm.

The online version of this article includes the following source data and figure supplement(s) for figure 1:

**Source data 1.** The numerical source data.

**Figure supplement 1.** The axon terminal enrichment of Wallenda (Wnd) is independent of Wnd kinase activity.

**Figure supplement 1—source data 1.** The numerical source data.

Next, we employed a transgene approach. We generated Wnd-GFP transgenes with and without *wnd* mRNA's untranslated region (UTR), namely 5'UTR-Wnd::GFP-3'UTR and Wnd::GFP to test the role of *wnd*-UTRs in Wnd protein localization. These transgenes were expressed specifically in larval class IV dendritic arborization (C4da) neurons using the *ppk*-GAL4 driver (*Grueber et al., 2007*). C4da neurons have their cell bodies positioned in the larval body wall, where they extend dendrites and send axons to the VNC of the larval central nervous system. This sensory neuron system provides an excellent opportunity for studying protein localization in cell bodies, dendrites, and axon terminals (*Figure 1B*; *Grueber et al., 2007*). We quantified the expression levels of Wnd transgenes in the cell bodies and axon terminals of C4da neurons using quantitative confocal microscope imaging. The expression levels were normalized using a general membrane marker, mCD8::RFP. The expression levels of mCD8::RFP in axon terminals may represent passive diffusion of membrane proteins, which reports the absence of axonal enrichment or equals to the axonal enrichment of 1. We observed a significant enrichment of Wnd in the axon terminals of C4da neurons, regardless of *wnd*-UTRs (*Figure 1C and D*). Overactive Wnd signaling impairs *ppk*-GAL4 activity, compounding our analysis (*Wang et al., 2013*). To address this, we treated larvae with DLKi/GNE-3511, a specific inhibitor of DLK/Wnd kinase activity (*Patel et al., 2017*; *Russo and DiAntonio, 2019*). The treatment with DLKi did not affect larval viability or basal Cd4a morphology. Additionally, a kinase-dead version of Wnd, Wnd-KD::GFP, exhibited similar axonal enrichment in comparison to Wnd::GFP in the presence and absence of DLKi (*Figure 1—figure supplement 1*).

What might be the biological significance of such high axon terminal enrichment of Wnd proteins? Interestingly, a previous work has shown that nerve injury in *Drosophila* larvae increased Wnd protein levels in proximal axons but not in motor neuron cell bodies (*Xiong et al., 2010*), which suggests that injury signal does not affect Wnd protein levels in neuronal cell bodies. Thus, we wondered whether Wnd protein degradation under normal conditions is also compartment-specific. To test this possibility, we expressed Wnd-KD::GFP in C4da neurons of wild-type and *hiw* mutant animals (*hiw[ΔN]*). The expression levels of Wnd-KD::GFP was measured in C4da cell bodies and C4da axon terminals by quantitative confocal microscope imaging (*Figure 1E*). The result showed that *hiw* mutations did not change Wnd-KD::GFP expression in C4da cell bodies as compared to a wild-type control. In contrast, there was a remarkable sevenfold increase in Wnd-KD::GFP expression levels specifically in the axon terminals in the presence of *hiw* mutations.

Taken together, these suggest that Wnd proteins are highly enriched in axon terminals and that this axon terminal localization is a necessary step for Hiw-mediated Wnd protein turnover. DLK proteins act as a sensor in neuronal stress responses, and under normal conditions, their protein levels are constantly suppressed to prevent undesired stress responses (*Nakata et al., 2005*; *Collins et al., 2006*; *Xiong et al., 2010*; *Huntwork-Rodriguez et al., 2013*). In this context, if Wnd localization in axon terminals is disrupted, it may lead to dysregulated Wnd protein expression levels, potentially resulting in exacerbated stress signaling. To test the model, we decided to identify the mechanisms underlying Wnd localization in axon terminals.

## The protein palmitoylation of Wnd is essential for axonal localization of Wnd

The coding region of *wnd* was sufficient for Wnd axon terminal localization (*Figure 1*). Wnd proteins are found on axonally transporting vesicles (*Xiong et al., 2010*). Together these further suggest that Wnd protein localization in axon terminals are through axonal anterograde transport of Wnd proteins, and that Wnd protein degradation is dependent on this axonal anterograde transport. Protein transport often involves small protein domains or motifs. To find the motifs necessary for the axonal transport of Wnd, we performed a structure-function analysis by progressively deleting 100 amino acids (aa) from Wnd transgenes (*Figure 2A*). These Wnd transgenes were expressed in C4da neurons, and the axonal enrichment of Wnd was measured as described in *Figure 1D*. Deletion of the first 100 aa of Wnd (WndΔ100::GFP) did not have an impact on the enrichment of Wnd in axon terminals. However, deleting the first 200 aa of Wnd (WndΔ200::GFP) resulted in a dramatic 43-fold reduction of Wnd enrichment. Similarly, deleting aa 101–200 of Wnd (WndΔ101–200::GFP) led to a dramatic decrease in Wnd localization in axon terminals (*Figure 2B*). These findings suggest that the 101–200 aa region of Wnd contains a crucial motif for the axonal anterograde transport of Wnd.

Interestingly, this region contains an evolutionarily conserved palmitoylation motif, Cysteine-130 (Wnd-C130) (*Figure 2C*). Previous studies have shown that palmitoylation is required for DLK association with neuronal vesicles and DLK signaling in cultured mammalian neurons and *C. elegans* (*Holland et al., 2016*). However, whether DLK palmitoylation changes DLK protein localization in axon terminals remains unknown. Additionally, whether Wnd undergoes protein palmitoylation is not known. To examine the protein palmitoylation of Wnd, we introduced a mutation changing the Cysteine to Serine, resulting in Wnd-C130S. We expressed wild-type Wnd (Wnd::GFP) and the palmitoylation site mutant Wnd (Wnd-C130S::GFP) in cultured *Drosophila* Schneider 2 (S2) cells using PEI-mediated transfection (*Longo et al., 2013*). The Acyl-Biotin Exchange (ABE) assay (*Drisdel et al., 2006*) was conducted to measure protein palmitoylation of Wnd proteins after pulldown using a GFP-tag. The results showed a 50% reduction in protein palmitoylation in Wnd-C130S::GFP compared to Wnd::GFP (*Figure 2D*), providing strong evidence that Wnd-C130 is a palmitoylation site. Subsequently, we expressed Wnd::GFP and Wnd-C130S::GFP in larval C4da neurons and measured their axonal enrichment. The findings revealed a dramatic 30-fold reduction in Wnd axonal localization for Wnd-C130S::GFP compared to Wnd::GFP (*Figure 2E*). Taken together, our data demonstrates that Wnd-C130 is palmitoylated in *Drosophila*, which is essential for Wnd localization in axon terminals.

Interestingly, we noticed that unlike diffuse cytoplasmic expression patterns of palmitoylation-defective DLK observed in mammalian cells (*Holland et al., 2016*), Wnd-C130S::GFP exhibited discrete localization patterns in C4da cell bodies (*Figure 2E*). These structures are extensively colocalized with somatic Golgi complex (*Figure 2—figure supplement 1*). In mammalian cultured cells, DLK was found in the somatic Golgi complex where its palmitoyltransferase, HIP14 localizes (*Niu et al., 2020*). Consistently, we found that Wnd is localized to somatic Golgi complex in C4da cell bodies, which required the 101–200 aa region of Wnd (*Figure 2—figure supplement 2*). *Drosophila* HIP14, dHIP14 showed somatic Golgi localization and extensive colocalization with Wnd proteins in C4da cell bodies (*Figure 2—figure supplement 3*). Furthermore, mutating *dHIP14* significantly decreased Wnd enrichment in axon terminals (*Figure 2—figure supplement 3*). These suggest that Wnd protein palmitoylation is catalyzed by dHIP14 in somatic Golgi complex, like mammalian cells. However, unlike mammalian and *C. elegans* DLK proteins, Wnd palmitoylation is not necessary for Golgi localization since Wnd-C130S::GFP retained Golgi localization (*Figure 2—figure supplement 1*). Moreover, Wnd palmitoylation plays a unique role in localizing Wnd proteins in axon terminals in *Drosophila*.

## Palmitoylation-defective Wnd evades protein turnover and accumulates in neuronal cell bodies

Wnd-C130S does not localize in the axon terminals but rather is retained in neuronal cell bodies. This may prevent protein turnover of Wnd-C130S, which may result in increased protein expression levels of Wnd-C130S. To test the idea, we expressed Wnd::GFP and Wnd-C130S::GFP using a pan-neuronal driver, *Elav*-GAL4. The larvae were treated with DLKi/GNE-3511 to prevent lethality caused by overactive Wnd signaling. The total protein expression levels of Wnd::GFP and Wnd-C130S::GFP were measured in larval brain lysates using western blot analysis with GFP antibody (*Figure 3A*). The results revealed a dramatic 7.1-fold increase in protein levels of Wnd-C130S::GFP compared

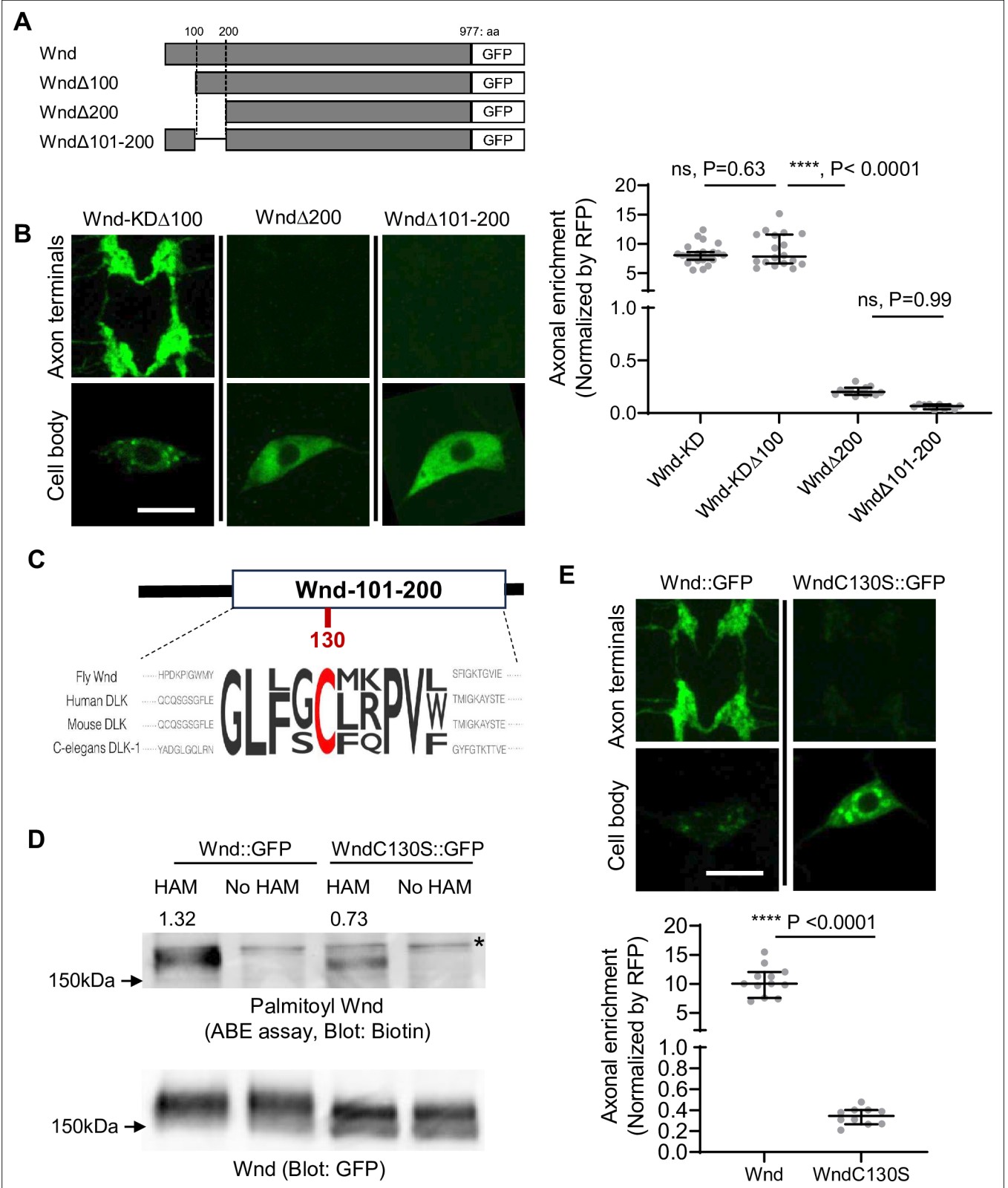

**Figure 2.** Wallenda (Wnd) palmitoylation is essential for Wnd axonal localization. (**A**) Schematic of Wnd deletion mutants. Deletion of the N-terminal 1-100aa of Wnd (WndΔ100), Deletion of the N-terminal 1–200 aa of Wnd (WndΔ200), and Deletion of the N-terminal 101–200 aa of Wnd (WndΔ101–200). (**B**) The transgenes of serially deleted Wnd (green) along with mCD8::mRFP were expressed in the larval C4da neurons. Note that WndΔ100 was further mutated to remove kinase activity (Wnd-KDΔ100). The expression levels of Wnd::GFP proteins in C4da cell body and the axon terminals were shown

*Figure 2 continued on next page*

*Figure 2 continued*

(left). Scale bars = 10 μm. The axonal enrichment of each Wnd transgenes was measured using mCD8::RFP (not shown in figure) as a normalization control and expressed as median ± 95% CI. One-way ANOVA (F(3,59) = 103.8) followed by post hoc Tukey's multiple comparison test. The p-values from Tukey's test are indicated in the graph. Genotypes and sample numbers were Wnd-KD (*UAS-Wnd-KD::GFP/+;ppk-Gal4, UAS-mCD8::RFP/+*, n=22), Wnd-KDΔ100 (*UAS-Wnd-KDΔ100::GFP/+;ppk-Gal4, UAS-mCD8::RFP/+*, n=18), WndΔ200 (*UAS-WndΔ200::GFP/+;ppk-Gal4, UAS-mCD8::RFP/+*, n=13), and WndΔ101–200 (*UAS-WndΔ101–200::GFP/+;ppk-Gal4, UAS-mCD8::RFP/+*, n=10). (**C**) Wnd-C130 is an evolutionarily conserved palmitoylation site. The protein sequence alignment of DLK proteins from the indicated species is shown. The conserved Cysteine residue is shown in red. (**D**) GFP-tagged Wnd proteins were expressed in S2 cells before being pulled down by GFP nanobody beads. The Acyl Biotin Exchange assay was performed on the beads coupled Wnd proteins. Wnd-C130S::GFP is a palmitoylation defective version of Wnd. 'No HAM' serves as a negative control since Biotin exchange does not occur in the absence of HAM. Biotin blot shows protein palmitoylation levels while total Wnd shows Wnd protein levels in the pulldown. The numbers indicate the total intensity of the palmitoylation blot that were normalized by GFP blot from Wnd::GFP and Wnd-C130S::GFP. (**E**) A wild-type Wnd::GFP transgene (Wnd::GFP) and a palmitoylation-defective Wnd transgene (Wnd-C130S::GFP) along with mCD8::RFP were expressed in the larval C4da neurons in the presence of DLKi. GFP immunostaining from Wnd::GFP proteins in C4da cell body and the axon terminals were shown (top). Scale bar = 10 μm. The axonal enrichment of the Wnd transgenes was measured using mCD8::RFP (not shown in figure) as a normalization control and expressed as median ± 95% CI (U=0, p<0.0001, two-tailed Mann-Whitney test). Genotypes and samples numbers were Wnd::GFP (*UAS-Wnd::GFP/+;ppk-Gal4, UAS-mCD8::RFP/+*, n=12) and Wnd-C130S::GFP (*UAS-WndC130S::GFP/+;ppk-Gal4, UAS-mCD8::RFP/+*, n=10).

The online version of this article includes the following source data and figure supplement(s) for figure 2:

**Source data 1.** The original western blots for *Figure 2D*.

**Source data 2.** The original western blots for *Figure 2D* with relevant bands labeled.

**Source data 3.** The numerical source data.

**Figure supplement 1.** Wallenda (Wnd) palmitoylation is not required for Wnd localization in somatic Golgi complex.

**Figure supplement 1—source data 1.** The numerical source data.

**Figure supplement 2.** Somatic Golgi localization is necessary for Wallenda (Wnd) localization in axon terminals.

**Figure supplement 2—source data 1.** The numerical source data.

**Figure supplement 3.** dHIP14 is required for Wallenda (Wnd) localization in axon terminals.

**Figure supplement 3—source data 1.** The numerical source data.

with Wnd::GFP. To determine the cellular localization of increased Wnd-C130S::GFP expression, we expressed Wnd::GFP and Wnd-C130S::GFP along with mCD8::RFP in larval C4da neurons using *ppk*-GAL4 and directly compared the protein expression levels of the Wnd transgenes in C4da axon terminals and cell bodies as measured by GFP fluorescent intensity (*Figure 3B*). We observed a dramatic 10-fold increase in protein expression levels of Wnd-C130S::GFP in C4da cell bodies compared with Wnd::GFP. On the other hand, Wnd-C130S::GFP exhibited greatly reduced protein expression levels in C4da axon terminals (threefold reduction) compared with Wnd::GFP. Together, these findings strongly support the notion that Wnd localization in axon terminals is crucial for protein turnover of Wnd.

## Palmitoylation-defective Wnd triggers excessive stress signaling and neuronal loss

DLK triggers neuronal stress signaling (*Nakata et al., 2005*; *Collins et al., 2006*; *Karney-Grobe et al., 2018*). A failure to suppress Wnd protein levels in Wnd-C130S::GFP may lead to excessive neuronal stress signaling. While the essential role of palmitoylation in DLK signaling has been demonstrated in *C. elegans* and cultured mammalian cells (*Holland et al., 2016*; *Martin et al., 2019*; *Niu et al., 2020*; *Niu et al., 2022*), its impact on the signaling capacity of Wnd remains unknown. In contrast to the diffuse cytoplasmic expression pattern of palmitoylation-deficient DLK proteins in worms and mammalian cells, Wnd-C130S::GFP exhibited clear somatic Golgi localization (*Figure 2—figure supplement 1*). A recent study suggested that the membrane localization of DLK proteins, rather than palmitoylation itself, is crucial for DLK signaling (*Tortosa et al., 2022*). Therefore, we investigated whether Wnd-C130S::GFP retains signaling capacity. We expressed Wnd::GFP and Wnd-C130S::GFP in larval C4da neurons using *ppk-GAL4*. We used *puc-lacZ*, a transcription reporter downstream of c-Jun N-terminal Kinase (JNK) as a proxy for Wnd signaling (*Martín-Blanco et al., 1998*). Expression levels of *puc-lacZ* were evaluated by immunostaining and its fluorescent intensity was measured in the C4da nucleus. The results showed robust induction of *puc-lacZ* by Wnd::GFP as well as by Wnd-C130S::GFP (*Figure 4—figure supplement 1A*). Additionally, the excessive axon arborization induced by Wnd signaling was also observed both in Wnd::GFP and Wnd-C130S::GFP-expressing

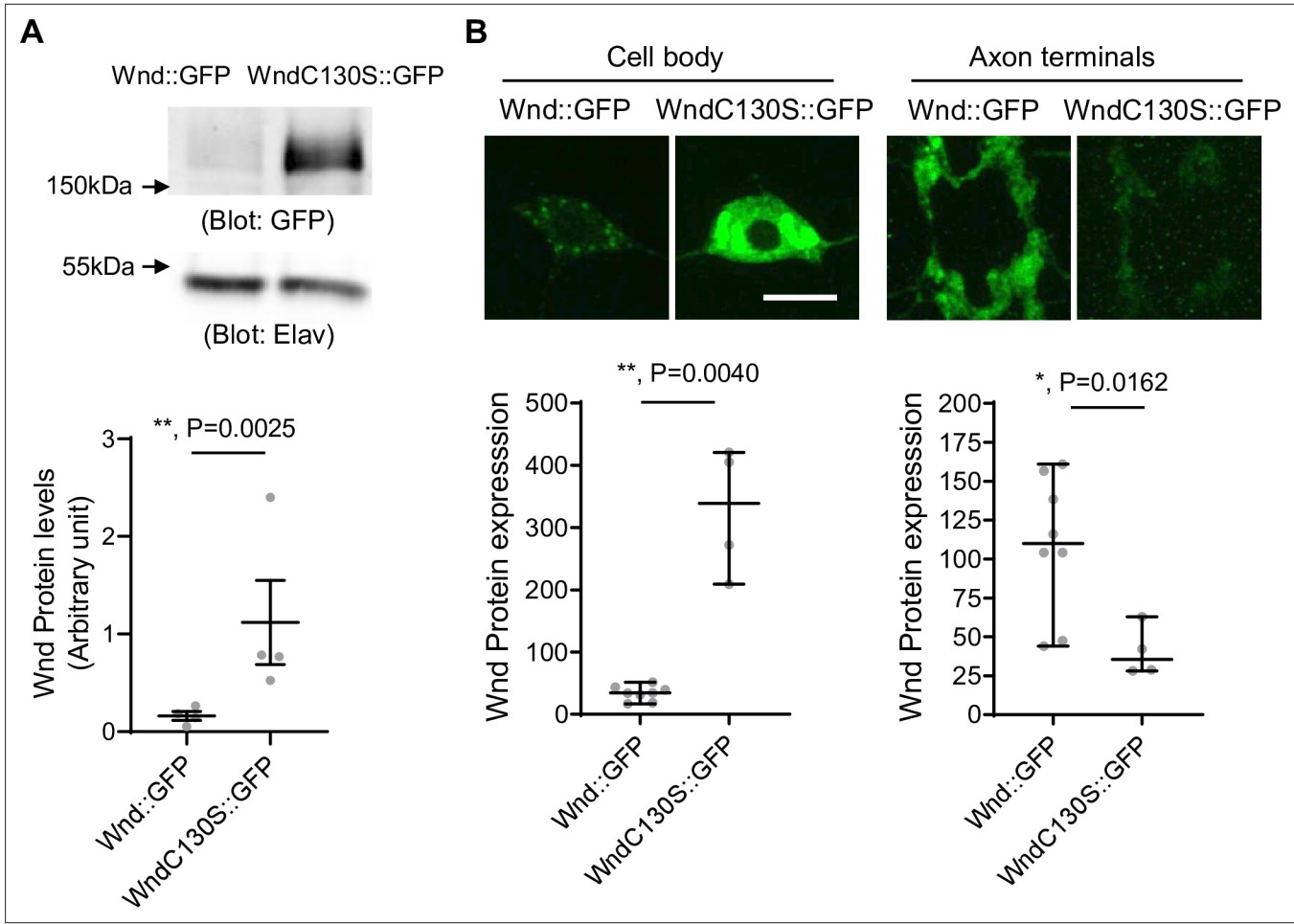

**Figure 3.** Increased protein expression levels of palmitoylation-defective Wallenda (Wnd). (**A**) Wnd::GFP and Wnd-C130S::GFP were expressed in the larval nervous system using a pan-neuronal driver, *Elav*-GAL4 in the presence of DLKi. Larval brain lysates were subjected to western blot analysis using a GFP antibody. Elav blot was used as a loading control. The GFP blots were normalized by Elav blots and expressed as mean ± SEM (n=4, t=9.409, df = 3, p=0.0025, two-tailed ratio paired t-test). Genotypes were Wnd::GFP (*elav-Gal4/+; UAS-Wnd::GFP/+;*) and WndC130S::GFP (*elav-Gal4/+; UAS-WndC130S::GFP/+;*). (**B**) Wnd transgenes (green) along with mCD8::RFP were expressed in the larval C4da neurons to directly compare the expression levels of Wnd::GFP and Wnd-C130S::GFP in C4da cell bodies and axon terminals. The larvae were treated with DLKi. Scale bar = 10 μm. The average fluorescent intensities from GFP staining of each larva sample (C4da cell bodies and axon terminals) were measured and expressed as median ± 95% CI (Cell body: Mann-Whitney U=0, p=0.0040, two-tailed Mann-Whitney test, Axon terminal: Mann-Whitney U=2, p=0.0162, two-tailed Mann-Whitney test,). Genotypes and sample numbers were Wnd::GFP (*UAS-Wnd::GFP/+;ppk-Gal4, UAS-mCD8::RFP/+*, n=8) and Wnd-C130S::GFP (*UAS-WndC130S::GFP/+;ppk-Gal4, UAS-mCD8::RFP/+*, n=4).

The online version of this article includes the following source data for figure 3:

**Source data 1.** The original western blots for *Figure 3A*.

**Source data 2.** The original western blots for *Figure 3A* with relevant bands labeled.

**Source data 3.** The numerical source data.

C4da neurons. These were dependent on Wnd kinase activity since there was no exuberant axon arborization by Wnd-KD::GFP nor by Wnd-C130S-KD::GFP – the kinase-dead versions of Wnd and Wnd-C130S, respectively (*Figure 4—figure supplement 1B*). These strongly suggest that, unlike DLK proteins in *C. elegans* and mammalian cells, the palmitoylation of Wnd is not essential for signaling in *Drosophila*.

Wnd-C130S::GFP exhibited a sevenfold increase in protein expression levels compared to Wnd::GFP (*Figure 3A*) while showing similar signaling capacity both in axon arborization and *puc-lacZ* expression. We wondered if excessive transgene expression levels caused signaling pathway saturation. Thus, we decided to reduce transgene expression levels. We recently have generated a plasmid

system for reduced transgene expression (*Singh and Kim, 2022*). Using these plasmids, additional Wnd transgenes containing two UAS sites (2XUAS) rather than five UAS sites (5XUAS) were generated and were expressed in larval C4da neurons. Then, *puc-lacZ* expression levels were measured. The result showed that Wnd-C130S::GFP induced a 3.7-fold more *puc-lacZ* expression than Wnd::GFP when expressed under 2XUAS (*Figure 4A*). This strongly suggests that Wnd-C130S::GFP triggers stronger stress signaling because it cannot be transported toward axon terminals thus, evades protein turnover.

To determine whether this excessive Wnd signaling from Wnd-C130S::GFP results in neuronal loss, we expressed Wnd-C130S::GFP and Wnd::GFP under *GMR-GAL4*, eye-specific driver (*Seong et al., 2001*). Expressing Wnd transgenes with 5XUAS caused complete lethality under a standard growth condition for both Wnd::GFP and Wnd-C130S::GFP. Growing flies at a lower temperature partially rescued lethality - likely due to lower transgene expression levels. Severe eye loss phenotypes were observed from Wnd-C130S::GFP and Wnd::GFP expressing flies in this experimental condition. We observed, albeit to a small extent, more severe eye loss phenotypes in Wnd-C130S::GFP flies as compared to Wnd::GFP flies (*Figure 4—figure supplement 2A and B*). To substantiate our findings, we expressed Wnd-C130S::GFP and Wnd::GFP under 2XUAS using *GMR-GAL4*. Even with 2XUAS transgenes, animals expressing Wnd-C130S::GFP showed significant lethality as compared to Wnd::GFP-expressing animals (*Figure 4B*). Moreover, 100% of surviving flies of Wnd-C130S::GFP exhibited rough eye and eye pigmentation loss phenotypes, which indicates loss of photoreceptors. On the other hand, flies expressing 2XUAS-Wnd::GFP did not show any discernable phenotypes as compared to the controls (*Figure 4C*). Wnd transgenes alone did not exhibit any defects in eye development.

These strongly suggest that axonal anterograde transport of Wnd is essential for Wnd protein turnover and that a failure of this transport triggers excessive stress signaling.

## Selective involvement of Rab proteins in Wnd axonal localization

Protein palmitoylation enhances the hydrophobicity of target proteins, leading to increased membrane association. This suggests that Wnd proteins are transported to axons through an intracellular vesicular transport system. The regulation of intracellular vesicular transport is governed by Rab proteins (*Stenmark, 2009*). To gain insights into the axonal anterograde transport of DLK proteins, we conducted a screening of Rab proteins to identify those that contribute to Wnd axonal localization. We tested a total of 26 Rab proteins out of the 33 annotated *Drosophila* Rab proteins (*Zhang et al., 2007*). YFP-tagged dominant-negative (DN) Rab proteins were expressed along with RFP-fused Wnd-KD (Wnd-KD::mRFP) in C4da neurons. The axonal enrichment of Wnd-KD::mRFP was analyzed by measuring the fluorescence intensities of Wnd-KD::mRFP in C4da cell bodies and axon terminals (*Figure 5*). Among the tested DN-Rab proteins, only Rab1 and Rab11 showed a significant reduction in Wnd axonal enrichment. These findings suggest that Wnd targeting to axon terminals is a highly specific process. Rab1 is involved in ER to Golgi traffic. The dramatic decrease in Wnd axonal localization observed upon Rab1 inhibition is interesting as Wnd palmitoylation likely occurs at the somatic Golgi complex. Rab11 regulates membrane traffic at the recycling endosome (*Stenmark, 2009*).

## Rab11 inhibition caused dysregulated Wnd protein levels

Rab11-positive recycling endosomes (REs) play a role in mediating anterograde Golgi-to-plasma membrane trafficking (*Ang et al., 2004*; *Satoh et al., 2005*; *Taguchi, 2013*). This raises an interesting possibility that once palmitoylated at the Golgi, Wnd is sorted at the REs for axonal transport. To investigate this, we examined whether Wnd proteins are present at the REs in C4da cell bodies. Wnd-KD::mRFP was expressed in C4da neurons, and the Rab11-positive REs were labeled using endogenously tagged YFP::Rab11 (TI-YFP::Rab11) (*Dunst et al., 2015*). We analyzed the portion of Wnd-KD::mRFP that colocalizes with YFP::Rab11 using the Manders' Overlap Coefficient. The results showed extensive colocalization between Wnd-KD::mRFP and YFP::Rab11 in C4da cell bodies (Manders' coefficient 0.60±0.14) (*Figure 6A*). Similar results were obtained with transgenic expression of YFP::Rab11 and Wnd-KD::mRFP (Manders' coefficient 0.81±0.11) (*Figure 6B*). On the other hand, we did not detect any meaningful colocalization between YFP::Rab11 and Wnd-KD::mRFP in C4da axon terminals or in axons (Manders' coefficient 0.34±0.14 and 0.41±0.10, respectively)

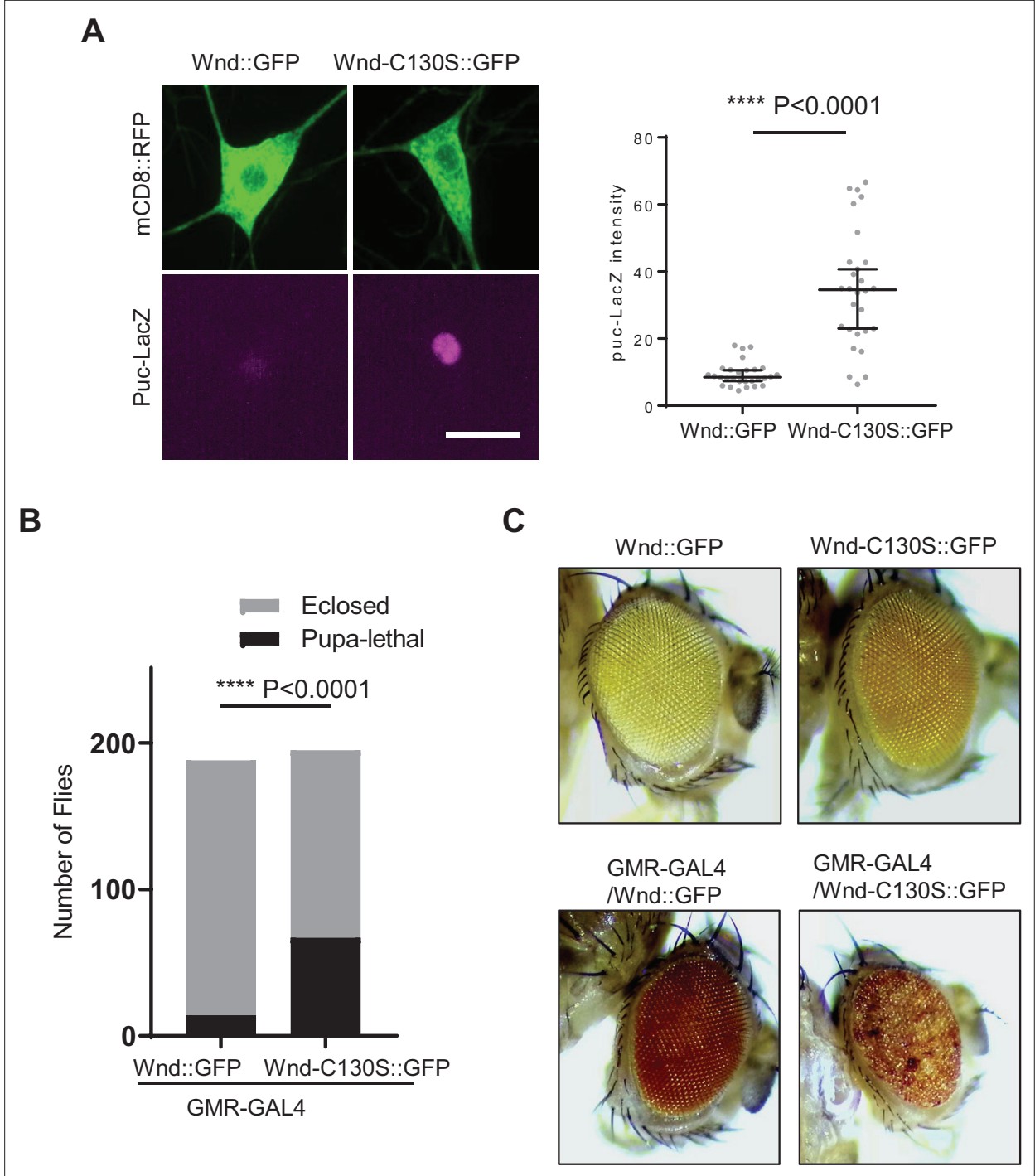

**Figure 4.** Palmitoylation-defective Wallenda (Wnd) caused exacerbated stress responses. (**A**) *puc*-LacZ was expressed along with *2XUAS-Wnd::GFP* and *2XUAS-Wnd-C130S::GFP* in the larval C4da neurons. The average fluorescent intensities from *puc*-LacZ staining of each larva sample in C4da cell body were measured and expressed as median ± 95% CI (Mann-Whitney U=56, p<0.0001, two-tailed Mann-Whitney test). Genotypes and sample numbers were 2XUAS-Wnd::GFP (*2XUAS-Wnd::GFP/ppk-Gal4,UAS-mCD8::RFP;puc-LacZ/+*, n=28) and 2XUAS-Wnd-C130S::GFP (*2XUAS-Wnd-C130S::GFP/ ppk-Gal4,UAS-mCD8::RFP;puc-LacZ/+*, n=28). Scale bar = 10 μm. (**B**) *2XUAS-Wnd::GFP* and *2XUAS-Wnd-C130S::GFP* were expressed under an eye-specific driver, *GMR-GAL4*. Adult fly lethality was scored and analyzed using the Chi-square (Chi-square, df = 41.57,1, z=6.448, p<0.0001, two-sided contingency test). Genotypes and sample numbers were Wnd::GFP (*2XUAS-Wnd::GFP/GMR-Gal4*, n=188), Wnd-C130S::GFP (*2XUAS-Wnd-C130S::GFP/GMR-Gal4*, n=195). (**C**) Representative images of fly eyes from Wnd::GFP (*2XUAS-Wnd::GFP/+*), Wnd-C130S::GFP (*2XUAS-Wnd-C130S::GFP/+*), Wnd::GFP under GMR-GAL4 (*2XUAS-Wnd::GFP/GMR-Gal4*), Wnd-C130S::GFP under GMR-GAL4 (*2XUAS-Wnd-C130S::GFP/GMR-Gal4*).

The online version of this article includes the following source data and figure supplement(s) for figure 4:

*Figure 4 continued on next page*

*Figure 4 continued*

**Source data 1.** The numerical source data.

**Figure supplement 1.** Wallenda (Wnd)-C130S::GFP retains Wnd signaling capacity.

**Figure supplement 1—source data 1.** The numerical source data.

**Figure supplement 2.** Wallenda (Wnd)-C130S::GFP exhibits higher toxicity levels than Wnd::GFP.

**Figure supplement 2—source data 1.** The numerical source data.

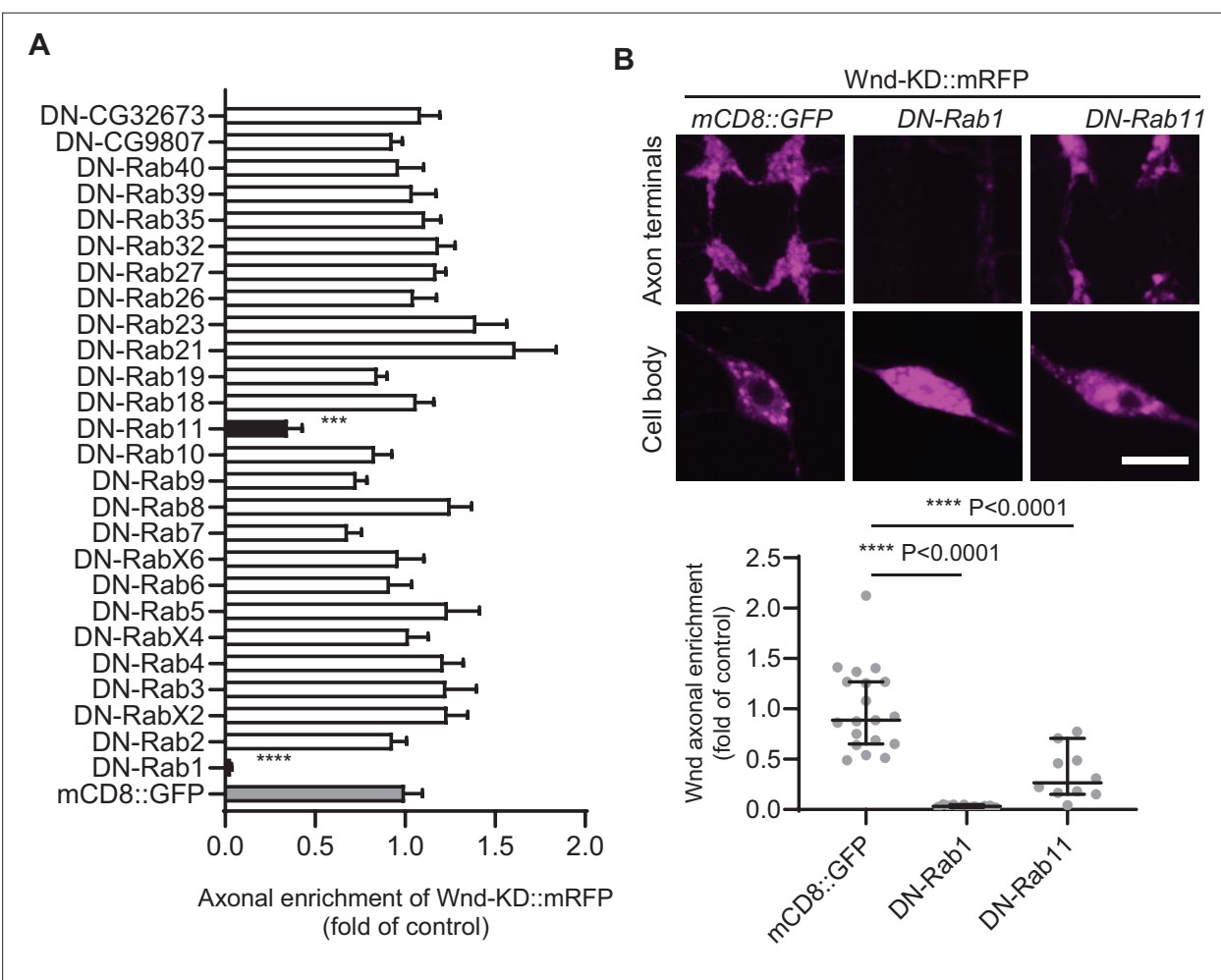

**Figure 5.** Screening of dominant negative Rab proteins for Wallenda (Wnd) axonal enrichment. (**A**) YFP-tagged dominant negative-Rab (DN-Rab) transgenes along with Wnd-KD::mRFP were expressed in the larval C4da neurons. The axonal enrichment of Wnd-KD::mRFP was measured by Wnd-KD::mRFP expression levels from C4da cell bodies and axon terminals and expressed as mean ± sem of a fold change of control (mCD8::GFP) (n≥10). One-way ANOVA ($F_{(27,284)} = 7.146$) followed by post hoc Dunnett test. A mCD8::GFP transgene was used as a transgene control (*UAS-Wnd-KD::mRFP/+;ppk-Gal4, UAS-mCD8::GFP/+*). Note that YFP::DN-Rab1 and YFP::DN-Rab11 reduced Wnd axonal enrichment by more than 50% compared to the control (gray bar). (**B**) Representative images of Wnd-KD::mRFP expression in the axon terminals and cell bodies from the C4da neurons that express mCD8::GFP, DN-Rab1, and DN-Rab11 were shown. The Wnd axonal enrichment was expressed as median ± 95% CI (Mann-Whitney U=0, p<0.0001 for DN-Rab1, U=13, p<0.0001 for DN-Rab11, two-tailed Mann-Whitney test). Genotypes and sample numbers were mCD8::GFP (*UAS-Wnd-KD::mRFP/+;ppk-Gal4, UAS-mCD8::GFP/+*, n=19) DN-Rab1 (*UAS-Wnd-KD::mRFP/+;ppk-Gal4 /UAST YFP.Rab1 S25N*, n=12) and DN-Rab11 (*UAS-Wnd-KD::mRFP/ UASp-YFP.Rab11 S25N; ppk-Gal4/+*, n=10) Scale bar = 10 μm.

The online version of this article includes the following source data for figure 5:

**Source data 1.** The numerical source data.

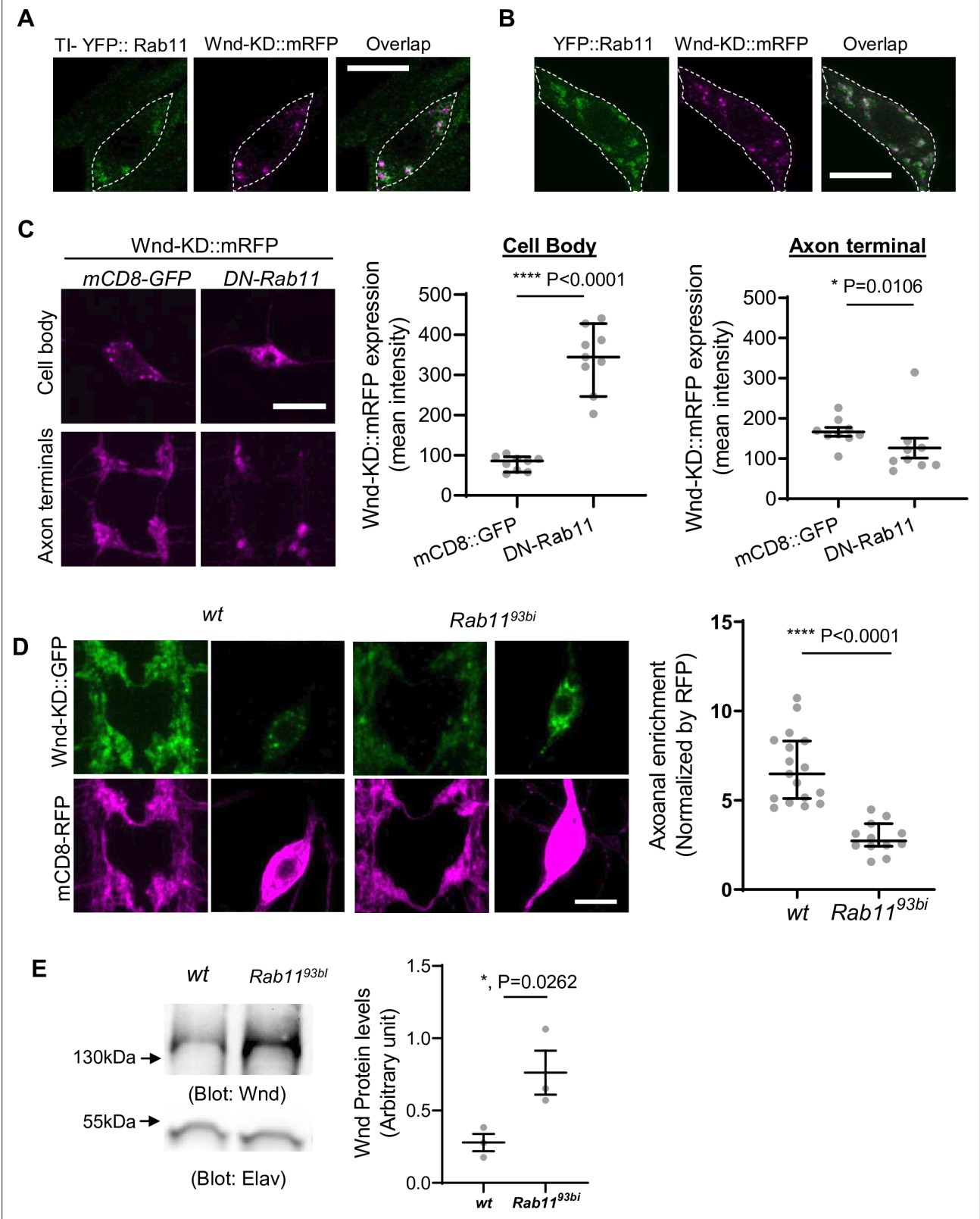

**Figure 6.** Rab11 is necessary for Wallenda (Wnd) localization in axon terminals and Wnd protein turnover. (**A**) and (**B**). Wnd localizes at the Rab11-positive endosomes in C4da cell bodies. Wnd-KD::mRFP was expressed in C4da neurons of *TI-YFP::Rab11* larvae (**A**). *TI-YFP::Rab11* is endogenously YFP-(Yellow fluorescent protein) tagged *Rab11* allele. Wnd-KD::mRFP was expressed in C4da neurons along with YFP::Rab11 (**B**). Wnd-KD::mRFP (magenta) and YFP::Rab11 (green) were visualized in C4da cell bodies using an RFP and GFP antibodies, respectively. Genotypes and sample numbers

*Figure 6 continued on next page*

Figure 6 continued

were *TI-YFP::Rab11* (*UAS-Wnd-KD::mRFP/+;TI{TI}Rab11[EYFP]/ppk-Gal4,* n=28) and *YFP::Rab11* (*UAS-Wnd-KD::mRFP/UAS-YFP::Rab11;ppk-Gal4/+,* n=30). Scale bar = 10 µm. (**C**) Wnd transgenes (magenta) along with mCD8::GFP or YFP::DN-Rab11 were expressed in the larval C4da neurons to directly compare the expression levels of Wnd-KD::mRFP in C4da cell bodies and axon terminals. Scale bar = 10 µm. The average fluorescent intensities from mRFP staining of each larva sample were measured and expressed as median ± 95% CI (Cell body: Mann-Whitney U=0, p<0.0001, two-tailed Mann-Whitney test; Axon terminal: Mann-Whitney U=12, p=0.0160, two-tailed Mann-Whitney test). Genotypes and sample numbers were *wt* (*UAS-Wnd-KD::mRFP/+;ppk-Gal4, UAS-mCD8::GFP/+,* n=9) and *YFP::DN-Rab11* (*UAS-Wnd-KD::mRFP/UASp-YFP.Rab11.S25N;ppk-Gal4, UAS-mCD8::RFP/+,* n=9). (**D**) Wnd-KD::GFP (green) along with mCD8::RFP (magenta) were expressed in the larval C4da neurons from wild-type (*wt*) or from homozygous *Rab11* mutants (*Rab11^93bi^*). Images from C4da axon terminals and cell bodies were shown. Scale bar = 10 µm. The axonal enrichment of Wnd-KD::GFP was measured using mCD8::RFP as a normalization control and expressed as median ± 95% CI (Mann-Whitney U=0, p<0.0001, two-tailed Mann-Whitney test). Genotypes and sample numbers were *wt* (*UAS-Wnd-KD::GFP/ppk-Gal4, U-mCD8::RFP,* n=17) and *Rab11^93bi^* (*UAS-Wnd-KD::GFP/ppk-Gal4, U-mCD8::RFP; Rab11^93bi^,* n=12). (**E**) A western blot was performed on the larval brain lysates from wild-type (*wt, w^1118^*) or homozygous *Rab11^93bi^* using Wnd antibody to measure total Wnd protein levels. Elav blot was used as a loading control. The Wnd blots were normalized by Elav blots and expressed as mean ± SEM (n=3, t=6.06, df = 2, p=0.0262, two-tailed ratio paired t-test).

The online version of this article includes the following source data and figure supplement(s) for figure 6:

**Source data 1.** The original western blots for *Figure 6E*.

**Source data 2.** The original western blots for *Figure 6E* with relevant bands labeled.

**Source data 3.** The numerical source data.

**Figure supplement 1.** Wallenda (Wnd) does not localize in the Rab11-positive endosomes in axons and axon terminals.

**Figure supplement 1—source data 1.** The numerical source data.

**Figure supplement 2.** *Rab11* mutations did not increase Wallenda (*wnd*) mRNA abundance.

**Figure supplement 2—source data 1.** The numerical source data.

(*Figure 6—figure supplement 1*). These suggest that Rab11 is involved in Wnd protein sorting at the somatic REs rather than transporting Wnd directly.

If Rab11 is necessary for Wnd anterograde transport, inhibiting Rab11 may lead to dysregulated Wnd protein turnover and increased Wnd protein levels. To test the idea, we expressed DN-YFP::Rab11 along with Wnd-KD::mRFP in C4da neurons. We directly compared the expression levels of Wnd-KD::mRFP in C4da cell bodies and axon terminals. A mCD8::GFP transgene was used as a control for DN-Rab11. The results revealed a dramatic 4.2-fold increase in Wnd-KD::mRFP expression levels in C4da cell bodies by DN-Rab11, but not in the axon terminals (*Figure 6C*), which is reminiscent of Wnd-C130S behavior (*Figure 3B*).

We further examined the role of Rab11 in Wnd axonal localization using *Rab11* mutants, *Rab11^93Bi^*, which is a strong loss-of-function allele (*Giansanti et al., 2007*). Wnd-KD::GFP was expressed along with mCD8::RFP in wild-type and homozygous *Rab11^93Bi^* C4da neurons, and the axonal enrichment of Wnd-KD::GFP was measured. The results showed a dramatic reduction in Wnd axonal enrichment by *Rab11* mutations (*Figure 6D*). Next, we measured the total expression levels of endogenous Wnd using western blot analysis on brain lysates from wild-type control and homozygous *Rab11^93Bi^* larvae (*Figure 6E*). The results showed a significant 2.8-fold increase in total Wnd protein levels by *Rab11* mutations. This increase in Wnd protein levels was not due to increased *wnd* gene expression because *wnd* mRNA levels were not increased by *Rab11* mutations (*Figure 6—figure supplement 2*). Rather, it showed a significant reduction in *wnd* mRNA, which may be caused by a possible negative feedback from excessive Wnd signaling. Together, these findings strongly suggest that Rab11 mediates Wnd protein transport out of neuronal cell bodies towards axon terminals and that a failure in this transport leads to dysregulated Wnd protein turnover.

## Neuronal loss caused by Rab11 loss of function requires Wnd activity

A previous work has shown that *Rab11* mutations caused photoreceptor loss in the *Drosophila* eye and JNK signaling activation (*Tiwari and Roy, 2009*). However, how *Rab11* mutations induce JNK signaling is not clear. One of the pathways leading to JNK activation is DLK/Wnd (*Collins et al., 2006*; *Xiong et al., 2010*; *Ghosh et al., 2011*). Thus, we hypothesized that increased Wnd expression contributes to the neuronal loss and JNK activation by *Rab11* loss-of-function. To test the idea, we expressed DN-Rab11 along with a membrane marker mCD8::RFP in the larval C4da sensory neurons using *ppk-GAL4* (*Grueber et al., 2007*). JNK activation was monitored by *puc-lacZ* induction

(*Martín-Blanco et al., 1998*). The result showed a 2.8-fold increase of *puc-lacz* expression in C4da neurons by DN-Rab11 (*Figure 7A*), which was completely abolished by treating larvae with DLKi/GNE-3511, Wnd kinase inhibitor. This strongly suggests that increased Wnd protein levels, hence Wnd activity mediates JNK activation induced by DN-Rab11. Expressing DN-Rab11 caused severe eye loss phenotypes (*Figure 7B*). To test whether increased Wnd expression contributes to the neuronal loss caused by DN-Rab11, we treated DN-Rab11 expressing flies with Wnd kinase inhibitor, DLKi/GNE-3511. The treatment partially but significantly mitigated eye loss phenotype caused by DN-Rab11 expression (Elav-GAL4/DN-Rab11) while having no effect on eye development in control flies (Elav-GAL4 and DN-Rab11) (*Figure 7B*). Furthermore, genetically reducing Wnd activity by introducing a loss of *wnd* allele, *wnd³*, in DN-Rab11 expressing photoreceptors significantly ameliorated the eye loss phenotypes by DN-Rab11 (*Figure 7C*).

Taken together, these support a model in which Wnd proteins are transported to the axon terminals for Hiw-mediated protein degradation under normal conditions, and that Rab11 is essential for anterograde sorting of Wnd towards axon terminals. In this model, Rab11 inactivation leads to defective anterograde transport of Wnd, thus a deficit in Wnd protein degradation. Subsequent increase in Wnd protein levels triggers excessive stress signaling leading to neuronal death (*Figure 8*).

## Discussion

Here, we report that axonal localization and protein turnover of Wnd are tightly coupled. A failure in this coupling leads to dysregulated Wnd protein turnover and triggers excessive stress signaling. We further uncover the role of Rab11 in this coupling.

Many neuronal proteins exhibit polarized protein expression patterns. DLK proteins are found in the axonal compartments of diverse animals (*Xiong et al., 2010*; *Klinedinst et al., 2013*; *Holland et al., 2016*; *Niu et al., 2020*; *Niu et al., 2022*). However, the extent of DLK axonal localization has not been thoroughly studied, presumably because of technical limitations. We employed *Drosophila* larval sensory neurons to answer this question. Our data suggests that Wnd proteins are highly enriched in the axon terminals of these neurons (*Figure 1*). Given that Wnd encodes a cytoplasmic protein, the extent of axonal enrichment was rather surprising. Wnd protein levels are constantly suppressed by Hiw in the axon terminals. Without such suppression, Wnd proteins would show well over 70-fold enrichment in axons. The UTRs of *wnd* are dispensable for Wnd axonal enrichment (*Figure 1C*), which suggests that axonal anterograde transport of Wnd is responsible for such axonal enrichment. This is supported by previous observations of DLK proteins on membrane vesicles undergoing both anterograde and retrograde transport in axons (*Xiong et al., 2010*; *Holland et al., 2016*; *Li et al., 2017*). Taken together, these indicate that Wnd proteins are synthesized in the neuronal cell body and subsequently transported into axon terminals through a highly active and efficient anterograde transport process.

What might be such a transport mechanism? DLK genes encode cytoplasmic proteins. Protein palmitoylation is an important post-translational mechanism for protein transport (*Aicart-Ramos et al., 2011*; *Tortosa and Hoogenraad, 2018*). It increases the hydrophobicity of target proteins to aid membrane association so that palmitoylated proteins are transported along with the membrane vesicles they associate. In our study, we found that protein palmitoylation on Wnd at the Cysteine residue 130 (Wnd-C130) is essential for its axonal localization. Interestingly, the palmitoylation-defective mutant Wnd-C130S exhibited large punctate structures in the cell body (*Figure 2E*), which colocalized with the somatic Golgi complex (*Figure 2—figure supplement 1*). This suggests that Wnd-C130 palmitoylation is not solely responsible for membrane association. The presence of residual palmitoylation in Wnd-C130S (*Figure 2D*) may indicate the existence of an additional palmitoylation site, which may explain how Wnd-C130S associates with the Golgi. Consistent with our findings, previous studies have shown that DLK proteins in mammalian cells and *C. elegans* are palmitoylated on a single Cysteine residue, and when this residue is mutated, the proteins exhibit a diffuse cytoplasmic expression pattern (*Holland et al., 2016*). These observations raise the question of why Wnd-C130S, with its residual palmitoylation and membrane association, fails to be transported to axon terminals. Interestingly, a recent research has proposed a role for protein palmitoylation in anterograde transport within the Golgi complex (*Ernst et al., 2018*). We envision that Wnd-C130 palmitoylation is a necessary sorting step towards the axonal anterograde transport from the somatic Golgi complex rather than merely acting as a membrane anchor.

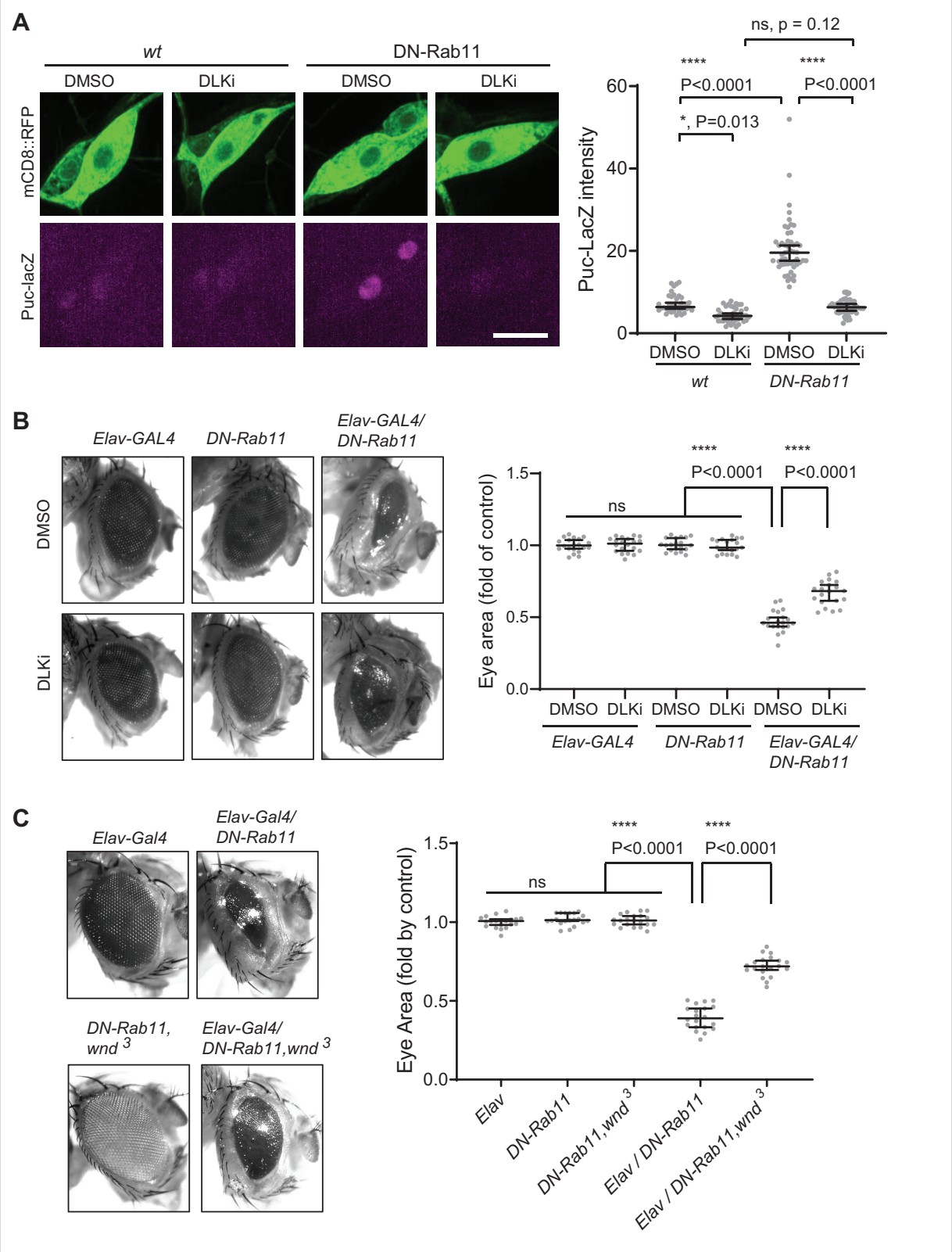

**Figure 7.** Wallenda (Wnd) mediates stress signaling induced by Rab11 loss-of-function. (**A**) *puc*-LacZ were expressed along with YFP::DN-Rab11 and mCD8::GFP in the larval C4da neurons in the presence of DMSO (vehicle) or dual leucine zipper kinase (DLK) inhibitor (DLKi). The average fluorescent intensities from LacZ staining of each larva sample in C4da cell bodies were measured and quantified as median ± 95% CI. One-way ANOVA (F (3171)=161.8) followed by post hoc Tukey's multiple comparison test (****p<0.0001). Genotypes and sample numbers were *wt (; ppk-Gal4, UAS-*

*Figure 7 continued on next page*

*Figure 7 continued*

*mCD8::RFP/+;puc-LacZ/+*, in DMSO, n=36, in DLKi n=42) and DN-Rab11 (*; ppk-Gal4, UAS-mCD8::RFP/+;puc-LacZ/UASp-YFP.Rab11.S25N*, in DMSO, n=51, in DLKi, n=46) Scale bar = 10 μm. (**B**) Representative images of fly eyes from *Elav-GAL4* only, *DN-Rab11* only, and *DN-Rab11* expression under a pan-neuronal driver, *Elav-GAL4* (*Elav-GAL4/DN-Rab11*). Flies were reared in the presence of DMSO and DLKi (down). Eye area was quantified and expressed as median ± 95% CI. One-way ANOVA (F (5,114)=290.3) followed by post hoc Tukey's multiple comparison test (****p<0.0001). Genotypes and sample numbers were Elav-Gal4 (*Elav-GAL4/+*, n=20), DN-Rab11 (*;;UASp-YFP.Rab11.S25N/+*, n=20), Elav-Gal4/DN-Rab11 (*ElavGal4/+;;UASp-YFP. Rab11.S25N /+*, n=20). (**C**) Representative images of fly eyes from *Elav-GAL4* only, *DN-Rab11* only in heterozygous *wnd* mutant (*DN-Rab11, wnd³*), DN-Rab11 expression under a pan-neuronal driver, *Elav-GAL4* (*Elav-GAL4/DN-Rab11*), and DN-Rab11 expression under a pan-neuronal driver in heterozygous *wnd* mutant (*Elav-GAL4/DN-Rab11, wnd³*). Eye area was quantified and expressed as median ± 95% CI. One-way ANOVA (F (495)=530.4) followed by post hoc Tukey's multiple comparison test.

The online version of this article includes the following source data for figure 7:

**Source data 1.** The numerical source data.

The role of Rab11 in Wnd axonal targeting and protein turnover uncovered by our DN-Rab screening suggests a role of RE in these processes (*Figures 5 and 6*). Rab11 plays a critical role in membrane trafficking events at the RE and is often used as a marker for the RE (*Taguchi, 2013*). It is involved in the anterograde transport of cargo from the Golgi apparatus to the plasma membrane through the RE (*Ang et al., 2004*; *Satoh et al., 2005*; *Taguchi, 2013*). We envision that once Wnd is palmitoylated, it is sorted at the peri-nuclear RE for subsequent axonal anterograde transport. Our observation of significant colocalization between Wnd proteins and Rab11-positive puncta in C4da cell bodies (*Figure 6*) but not in axon terminals or in axons (*Figure 6—figure supplement 1*) supports the idea that Wnd proteins are associated with the RE during their anterograde transport. These further suggest that Rab11 is not directly involved in the anterograde long-distance transport of Wnd proteins, but rather is responsible for sorting Wnd into the axonal anterograde transporting vesicles.

Inhibiting Rab11 activity led to increased Wnd protein levels (*Figure 6E*). Reduced activity or expression levels of Rab11 have been reported in experimental models of Huntington's disease, Parkinson's disease and in patients with amyotrophic lateral sclerosis (*Li et al., 2009a*, *Li et al., 2010*; *Breda et al., 2015*; *Mitra et al., 2019*; *Rai and Kumar Roy, 2022*). Furthermore, recent studies point to the involvement of DLK in multiple neurodegenerative conditions (*Pozniak et al., 2013*; *Welsbie et al., 2013*; *Welsbie et al., 2017*; *Larhammar et al., 2017a*; *Le Pichon et al., 2017*; *Patel et al., 2017*). These raise a tantalizing possibility that disrupted anterograde transport of DLK, and

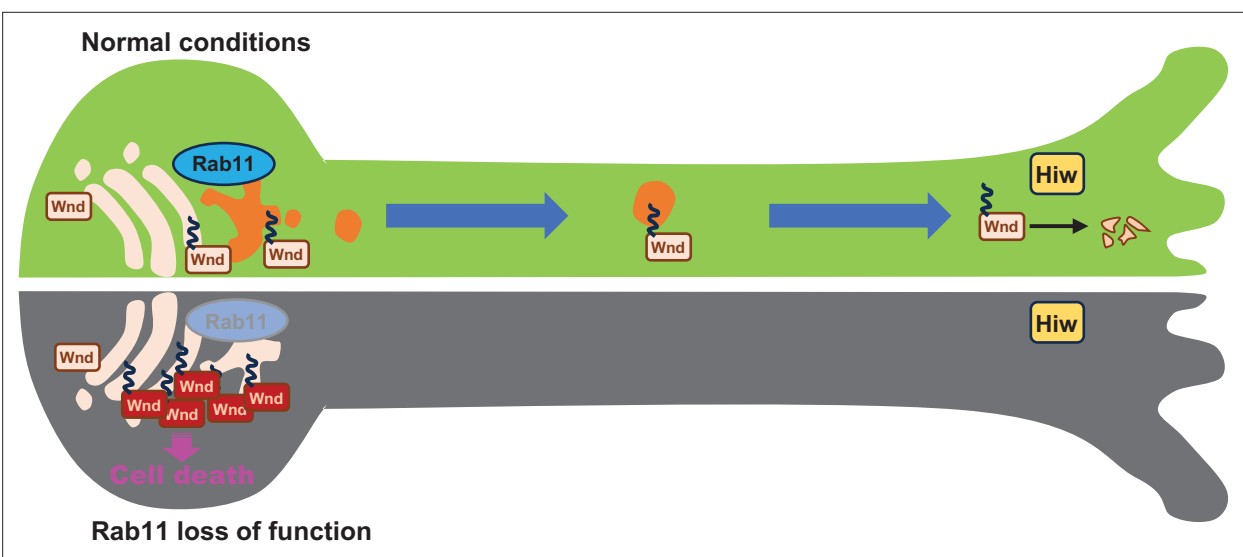

**Figure 8.** Rab11 suppresses neuronal stress signaling by localizing Dual leucine zipper kinase to axon terminals for protein turnover. Under normal conditions, the newly synthesized Wallenda (Wnd) proteins are recruited to the Golgi apparatus in the neuronal cell body. Wnd is palmitoylated at somatic Golgi apparatus, which enables it to be sorted into the Recycling Endosomes via Rab11. Wnd is actively transported out of neuronal cell bodies towards axon terminals where Highwire (Hiw)-mediated degradation of Wnd occurs. When Rab11 function is lost, Wnd proteins cannot be transported towards axon terminals. This prevents Wnd protein turnover resulting in higher Wnd protein expression, which in turn triggers the cell death pathway.

subsequent dysregulated DLK protein turnover, could be a contributing factor in these neurodegenerative disorders. In this regard, it is highly encouraging that inhibiting Wnd activity mitigated the neuronal loss and JNK activation by *Rab11* loss-of-function (*Figure 7*). Further investigations are needed to elucidate the causal role of dysregulated DLK protein turnover in neurodegenerative conditions and to better understand the interplay between Rab11-mediated transport, protein turnover, and DLK-mediated stress signaling.

The mechanism by which Hiw performs compartmentalized Wnd protein turnover in axons is an interesting question. There are several possibilities to consider. One possibility is that Hiw protein itself is specifically present in the axon terminals and not in the cell body. Studies have shown that Phr1, the mammalian homolog of Hiw, is expressed in axonal segments of motor neuron explants and in the axon shaft of sensory neurons, but its expression in the growth cone is low or absent (*Lewcock et al., 2007*). However, it is not known whether Hiw proteins are present in the cell body and to what extent. Unfortunately, the lack of an immunostaining compatible Hiw antibody has limited our ability to test this possibility. Nevertheless, transgenic expression of GFP::Hiw in *Drosophila* has shown significantly higher expression in the motor neuron cell bodies (*Xiong et al., 2010*). Moreover, a study has demonstrated that Hiw controls the expression of dNMNAT2, a substrate of Hiw, both in the cell body and the neuromuscular junction (*Xiong et al., 2012*). These observations suggest that Hiw expression itself may not be restricted to the axon terminals. Another possibility is that DLK proteins are protected from PHR1/RMP-1/Hiw-mediated protein turnover in the cell body. DLK protein stability can be enhanced by Hsp90 (*Karney-Grobe et al., 2018*), protein phosphorylation by JNK (*Huntwork-Rodriguez et al., 2013*), and PKA (*Hao et al., 2016*). These protective pathways may collaborate with PHR1/RMP-1/Hiw to execute axon-specific protein turnover of DLK proteins. Understanding the interplay between these protective mechanisms and PHR1/RMP-1/Hiw in regulating DLK protein turnover will provide valuable insights on subcellular-specific Wnd protein turnover.

The coupling between axonal transport and protein turnover of DLK proteins may have important implications in the context of neurodegenerative disorders. Recent studies have implicated DLK as a central player in neuronal death, including in diseases such as Alzheimer's disease and amyotrophic lateral sclerosis (*Pozniak et al., 2013*; *Welsbie et al., 2013*; *Welsbie et al., 2017*; *Larhammar et al., 2017a*; *Larhammar et al., 2017b*; *Le Pichon et al., 2017*; *Patel et al., 2017*). Defective axonal transport is a common feature observed in many neurodegenerative disorders (*Chevalier-Larsen and Holzbaur, 2006*; *De Vos et al., 2008*; *Guo et al., 2020*), and it has been proposed to contribute to disease pathogenesis. Interestingly, studies have reported increased DLK protein levels in neuronal cell bodies following neuronal insults (*Watkins et al., 2013*; *Welsbie et al., 2013*). Furthermore, DLK signaling has been shown to be activated by cytoskeletal disruptions (*Valakh et al., 2013*; *Valakh et al., 2015*), which may impair cytoskeleton-based axonal transport. This raises an intriguing possibility that disrupted axonal transport, leading to the accumulation of DLK proteins, could be a contributing factor for DLK activation and neuronal loss in neurodegenerative diseases.

## Materials and methods

### *Drosophila* strains, rearing, DNA constructs for generating transgenic flies

The following *Drosophila* strains were reared at 25 °C in a humidified chamber. *ppk-GAL4* (*Grueber et al., 2007*); *UAS-dHIP14::Tdtomato* and *Dhip14^ex11* (*Stowers and Isacoff, 2007*); *hiw^AN* (*Wu et al., 2005*); *UAS-ManII::GFP* (*Ye et al., 2007*); *UAS-ManII-Tag::RFP* (*Zhou et al., 2014*); *Mi{MIC}wnd ^GFSTF* (*Venken et al., 2011*); *GMR-Gal4* (*Seong et al., 2001*); *elav-Gal4^C155* (BDSC #458); *w^1118* (BDSC #3605); *TI{TI}Rab11^EYFP* (BDSC #62549) (*Dunst et al., 2015*); *UASp-YFP.Rab11* (BDSC #50782) (*Zhang et al., 2007*); *Rab11^93Bi* (BDSC #4158) (*Giansanti et al., 2007*); *puc-lacZ* (*Martín-Blanco et al., 1998*). For DN-Rab screening, the fly lines (*Zhang et al., 2007*) were obtained from the Bloomington *Drosophila* Stock Center and the full list of fly lines is provided in the Key Resources Table (Appendix 1—key resources table).

To generate DNA constructs of *Wnd* with its natural 5'- and 3'-UTRs and the serial deletion constructs of Wnd, the genomic sequences of the 5'-UTR and 3'-UTR of *Wnd* were obtained from *w^1118* genomic DNA by PCR. The coding sequence of *Wnd* was recovered from UAS-Wnd (*Collins et al., 2006*) via PCR and inserted into the pUASTattB plasmid using NotI and XbaI restriction sites.

UAS-Wnd-KD::GFP, Wnd-KDΔ100::GFP, and WndC130S-KD::GFP were generated by site-directed mutagenesis using In-Fusion Snap Assembly Master Mix (Takara Bio USA, Inc, San Jose, CA). UAS-Wnd-C130S::GFP and UAS-WndΔ101–200::GFP were generated by site-directed mutagenesis. To generate UAS-Wnd-KD::mRFP, the coding sequence of Wnd-KD was obtained from UAS-Wnd-KD::GFP via PCR and inserted into the pUASTattB plasmid using NotI and XbaI sites. To generate 2XUAS-Wnd::GFP and 2XUAS-Wnd-C130S::GFP, the coding sequences of Wnd and Wnd-C130S were obtained from UAS-Wnd::GFP and UAS-Wnd-C130S::GFP via PCR, respectively. Subsequently, they were inserted into the p2xUASTattB plasmid (Addgene #189860) (*Singh and Kim, 2022*) using NotI and XbaI restriction sites.

Transgenic flies were generated by PhiC31-mediated germline transformation using an identical landing site (*Bischof et al., 2007*).

To dose larvae with DLKi (GNE-3511, Sigma Aldrich), DLKi was dissolved in DMSO and was added to a final concentration of 35 µM in fly food. Fly larvae were raised directly on either control (DMSO) or DLKi containing food.

## *Drosophila* Schneider 2 cell culture, transfection, palmitoylation assay, and western blot

*Drosophila* Schneider 2 (S2) cells were prepared as previously described (*Singh et al., 2022*). *Drosophila* S2R + cells were purchased from the *Drosophila* RNAi Screening Center at Harvard University. The cells were amplified through a minimum number of passages (typically fewer than five). The absence of mycoplasma infection was confirmed by fluorescent microscopy using DAPI staining before aliquoting and storing in liquid nitrogen storage. Cells were visually inspected in each passage for any morphological alterations under a light microscope for quality control. The cells were discarded after the passage-number of 30. Cells were transfected with indicated DNA constructs and tubulin-Gal4 (*Lee and Luo, 1999*) using polyethyleneimine (Polysciences, Warrington, PA) (*Longo et al., 2013*).

Two days after transfection, the ABE assay was performed (*Drisdel et al., 2006*). Cells were lysed in the ice-cold Lysis buffer (LB, 50 mM HEPES/pH 7.4, 150 mM NaCl, 1 mM EDTA, 1% NP40, 5% glycerol, 1 mM PMSF, 1 mM $Na_3VO_4$, 5 mM NaF) with Halt protease inhibitor cocktail EDTA- free (Thermo Fisher Scientific, Waltham, MA) and 25 mM N-ethylmaleimide (Sigma, St. Louis, MO), incubated for 1 hr at 4 °C and centrifuged at 1000 × g for 5 min. The resulting post-nuclear lysates were centrifuged at 20,000 × g for 30 min at 4 °C. The supernatant was incubated with the GFP selector (NanoTag Biotechnologies, Göttingen, Germany) for 2 hr at 4 °C to pulldown Wnd-GFP proteins. The GFP selector beads along with bound proteins were recovered by a brief centrifugation and washed two times with LB containing 0.1% SDS before incubating with 1 M hydroxylamine hydrochloride (HAM, Sigma, St. Louis, MO) in LB (pH 7.2) at room temperature for 1 hr. Half of the protein-bound GFP selector beads were incubated with LB without HAM and used as a '- HAM' control. Then, the protein GFP selector beads were washed with pH 6.2 LB twice and treated with 1.6 µM EZ-Link BMCC-Biotin (Thermo Fisher Scientific, Waltham, MA) in pH 6.2 LB for 45 min at 4 °C. Beads were washed twice with LB without NP40 before being processed for SDS-PAGE. Proteins were separated by SDS-PAGE and were subjected to western blot analysis with mouse monoclonal anti-Biotin antibody (Jackson Immunoresearch, West Grove, PA) and chicken polyclonal anti-GFP antibody (Aves Labs, Tigard, OR).

To measure Wnd expression in larva brain lysates, larvae were grown on fly food containing 35 µM DLKi in a humidified chamber. The larval central nervous system (the ventral nerve cord and brain lobes) was dissected out from the third instar larvae and collected in ice-cold PBS before recovery by a brief centrifugation at 4 °C. The total brain lysates were prepared by directly adding SDS-PAGE sample buffer and homogenization before being resolved on 8% SDS-PAGE gel and subsequent western blot analysis. The samples were blotted using an anti-Wnd antibody (*Collins et al., 2006*) to estimate total Wnd proteins. Anti-Elav antibody (DSHB - 9F8A9) (*O'Neill et al., 1994*) was used as a loading control.

## Immunostaining and imaging

*Drosophila* third instar larvae were prepared for immunostaining as previously described (*Singh et al., 2022*). Primary antibodies used were rabbit polyclonal anti-RFP (Rockland Immunochemicals, Limerick, PA), chicken polyclonal anti-GFP (Aves Labs, Tigard, OR), and goat anti-horseradish peroxidase conjugated with Cy5 (Jackson ImmunoResearch, West Grove, PA). The secondary antibodies

used were Cy2- or Cy5-conjugated goat anti-chicken and Cy2- or Cy5-conjugated goat anti-Rabbit (Jackson ImmunoResearch, West Grove, PA).

Confocal imaging was done with a Leica SP8 confocal system, or a custom-built spinning disk confocal microscope equipped with a 63 x oil-immersion objective with a 0.3 µm step-sizes.

For Wnd axonal enrichment analysis, the C4da neuron cell bodies from abdominal segments 4–6 and the corresponding segments in the ventral nerve cord were imaged. The resulting 3D images were projected into 2D images using a maximum projection method in the ImageJ software. A region of interest was drawn in the cell body of C4da neurons, and the mean fluorescence intensity was measured using ImageJ software. The mean fluorescence intensity of UAS-mCD8::mRFP transgene expression was measured from the same region of interest and used as a normalization control. The mean intensities of the C4da cell bodies from a single larva were averaged out to calculate the axonal enrichment index. The axonal enrichment index was defined as Wnd expression (mean intensity) in the axon terminals divided by an average Wnd expression in cell bodies (mean intensity).

To analyze Golgi and Wnd colocalization, the resulting 3D images were projected into 2D images using a maximum projection method in the ImageJ software. The Manders' Overlap Coefficient (*Pike et al., 2017*) was measured as the percentage of total GFP signal overlapping with the RFP signal using the ImageJ plugin, JACoP (*Bolte and Cordelières, 2006*). To analyze puncta colocalization, we used the Manders' coefficient (*Pike et al., 2017*) that is a correlation between each channel signal using JACop. For Wnd colocalization with TI-YFP::Rab11, a single confocal slice was selected.

## *Drosophila* eye degeneration

The flies that are 24–48 hr after eclosion were imaged using the Basler ace acA2440-75 µm CCD-camera or Moticam A5 camera installed on a stereomicroscope. The region of interest was drawn on the fly eye area from the images and expressed as pixel numbers divided by 1000. Flies were raised at 27 °C in a humidified incubator except for *Figure 4—figure supplement 2*.

## *Drosophila* viability and lethality analysis

Flies were raised at 27 °C. Viability was determined by counting the number of filial pupae generation (adult flies after eclosion) and pupae lethality by counting the number of dead pupae.

## Quantitative reverse transcription PCR

Total RNAs were extracted from the wondering third instar larval brains from a wild-type ($w^{1118}$) or homozygous *Rab11* $^{93bi}$ mutants using the Quick-RNA MicroPrep kit (Zymo Research, Irvine, CA), and were subjected to reverse transcription using a mixture of gene-specific primers as follows, GAPDH1 (CTTCATTCGATGCACAAGTTTTATTTTTCAAATAGCTGG) and wnd (CCATACAGTTTAGGTCGATT TCTC, CAACTTCGGGCCTAATCAAGG, GCGTAACTTGGACAACTCCAA, GGAGATAGTTTTTGCT GCGTAAC, and GGCCCCCACCTAGACTAGAT).

The quantitative PCR was conducted using SYBR MasterMix (Thermo Fisher Scientific, Waltham, MA). The following gene-specific primer pairs were used.

GAPDH1, ATGTCTCCGTTGTGGATCTTAC and CCTCGACCTTAGCCTTGATTT; wnd primer1, TCTGCGGAGAAGAAGCATTGG, and TGGATGGCAATCTTCTGGGAG; wnd primer 2, GTTCGATA GGCAATGCCAAC and CGCAGAGCGTACTTGTAGTC.

The mRNA abundance of *wnd* was calculated by the 2ΔΔCT method using GAPDH1 as a normalizing control.

## Experimental design and statistical analysis

All statistical analysis was performed as two-tailed using GraphPad Prism software version 7.04. The Mann–Whitney test was used for immunostaining results. A ratio-paired t-test was used for western blot analysis (*Figures 3A and 6E*). For multigroup comparisons, one-way ANOVA was used followed by post hoc Tukey's multiple comparisons tests. The Chi-square test was performed

for *Drosophila* lethality analysis (*Figure 4B*). A p-value smaller than 0.05 was considered statistically significant. All p-values are indicated as NS; non-significant, *p<0.05, **p<0.01, ***p<0.001, and ****p<0.0001.

## Acknowledgements

We thank Drs. Steve Stowers and Catherine Collins for sharing reagents. Research reported in this study used the Cellular and Molecular Imaging Core facility at the University of Nevada Reno, which was supported by NIH P30 GM145646. This study was supported by NIH R01NS116463 and P20GM103440 (NV INBRE) to JK. The content is solely the responsibility of the authors and does not necessarily represent the official views of the National Institutes of Health.

## Additional information

### Funding

| Funder | Grant reference number | Author |
|---|---|---|
| National Institute of Neurological Disorders and Stroke | R01 NS116463 | Seung Mi Kim Yaw Quagraine Monika Singh Jung Hwan Kim |
| National Institute of General Medical Sciences | P30 GM145646 | Seung Mi Kim Yaw Quagraine Monika Singh Jung Hwan Kim |
| National Institute of General Medical Sciences | P20 GM103440 | Seung Mi Kim Yaw Quagraine Monika Singh Jung Hwan Kim |

The funders had no role in study design, data collection and interpretation, or the decision to submit the work for publication.

### Author contributions

Seung Mi Kim, Conceptualization, Data curation, Formal analysis, Supervision, Validation, Investigation, Methodology, Writing - original draft, Writing - review and editing; Yaw Quagraine, Formal analysis, Validation, Investigation, Methodology, Writing - original draft; Monika Singh, Formal analysis, Investigation, Methodology, Writing - original draft; Jung Hwan Kim, Conceptualization, Data curation, Formal analysis, Supervision, Funding acquisition, Investigation, Writing - original draft, Project administration, Writing - review and editing

### Author ORCIDs

Jung Hwan Kim 🆔 https://orcid.org/0000-0001-8548-4435

Reviewer #1 (Public Review): https://doi.org/10.7554/eLife.96592.3.sa1
Reviewer #2 (Public Review): https://doi.org/10.7554/eLife.96592.3.sa2
Author response https://doi.org/10.7554/eLife.96592.3.sa3

## Additional files

### Supplementary files
• MDAR checklist

### Data availability
All data generated or analysed during this study are included in the manuscript and supporting files.

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

## Appendix 1

### Appendix 1—key resources table

| Reagent type (species) or resource | Designation | Source or reference | Identifiers | Additional information |
|---|---|---|---|---|
| Antibody | anti-GFP (chicken polyclonal) | Aves Labs | Cat# GFP-1010, RRID: AB_2307313 | IF (1:250), WB (1:1000) |
| Antibody | anti-RFP (rabbit polyclonal) | Rockland | Cat# 600-401-379, RRID: AB_2209751 | IF (1:250) |
| Antibody | anti-LacZ (mouse monoclonal) | Developmental Studies Hybridoma Bank | Cat# 40–1 a | IF (1:5) |
| Antibody | Cy2- conjugated AffiniPure Donkey anti-chicken | Jackson ImmunoResearch | Cat# 703-225-155, RRID:AB_2340370 | IF (1: 500) |
| Antibody | Cy2-conjugated AffiniPure Donkey anti-rabbit | Jackson ImmunoResearch | Cat# 711-225-152, RRID:AB_2340612 | IF (1: 500) |
| Antibody | Cy2-conjugated AffiniPure Donkey anti-mouse | Jackson ImmunoResearch | Cat# 715-225-151, RRID:AB_2340827 | IF (1: 500) |
| Antibody | Cy5-conjugated AffiniPure Donkey anti-rabbit | Jackson ImmunoReesearch | Cat# 711-175-152, RRID:AB_2340607 | IF (1: 500) |
| Antibody | Cy5-conjugated AffiniPure Donkey anti-mouse | Jackson ImmunoResearch | Cat# 715-175-151, RRID:AB_2340820 | IF (1: 500) |
| Antibody | Cy5-conjugated AffiniPure Donkey anti-chic | Jackson ImmunoResearch | Code# 703-175-155, RRID:AB_2340365 | IF (1: 500) |
| Antibody | anti-Biotin (mouse monoclonal) | Jackson ImmunoResearch | Cat# 200-002-211, RRID: AB_2339006 | WB (1:1000) |
| Antibody | anti-Elav (rat monoclonal) | Developmental Studies Hybridoma Bank | Cat# Rat-Elav-7E8A10 anti-elav, RRID: AB_528218 | WB (1:10000) |
| Antibody | anti-Wnd (rabbit polyclonal) | PMID: 16815332 | Gift from Catherine A Collins | WB (1:1000) |
| Chemical compound, drug | GNE-3511 | Sigma-Aldrich | Cat# 5331680001 | DLK inhibitor |
| Chemical compound, drug | Halt protease inhibitor cocktail EDTA- free(100 X) | Thermo Fisher Scientific | Cat# 78437 | |
| Chemical compound, drug | N-ethylmaleimide | Sigma-Aldrich | Cat #04260 | |
| Chemical compound, drug | Hydroxylamine hydrochloride | Sigma-Aldrich | Cat# 55460 | |
| Chemical compound, drug | PEI | Polyscience | Cat# 23966 | |
| Peptide, recombinant protein | EZ-Link BMCC-biotin | Thermo Fisher Scientific | Cat# 21900 | |
| Other | Opti-MEM I | Gibco | Cat# 31985070 | Transfection reagent |
| Other | *Drosophila* Schneider's Medium | Thermo Fisher Scientific | Cat# 21720024 | Cell culture medium |
| Other | Fetal bovine serum, Heat Inactivated | Sigma-Aldrich | Cat # F4135 | Cell culture medium |
| Commercial assay or kit | GFP Selector affinity resin | Nano tag | Cat #N0310 | |
| Commercial assay or kit | In-Fusion HD Cloning | Takara Bio USA, Inc | Clontech:639647 | |
| Cell line (*D. melanogaster*) | *Drosophila* Schneider 2 (S2)-DRSC | *Drosophila* Genomics Resource Center | Cat# 181, RRID: CVCL_Z992 | |
| Software, algorithm | ImageJ | National Institutes of Health | RRID:SCR_003070 | |
| Software, algorithm | Fiji | National Institutes of Health | RRID:SCR_002285 | |

*Appendix 1 Continued on next page*

*Appendix 1 Continued*

| Reagent type (species) or resource | Designation | Source or reference | Identifiers | Additional information |
|---|---|---|---|---|
| Software, algorithm | GraphPad Prism | GraphPad | RRID:SCR_002798 | |
| Strain, strain background (*Drosophila melanogaster*) | UAST-YFP.Rab1.S25N | Bloomington *Drosophila* Stock Center | BDSC_9757 | |
| Strain, strain background (*Drosophila melanogaster*) | UAST-YFP.Rab2.S20N | Bloomington *Drosophila* Stock Center | BDSC_23640 | |
| Strain, strain background (*Drosophila melanogaster*) | UASp-YFP.RabX2.S21N | Bloomington *Drosophila* Stock Center | BDSC_9843 | |
| Strain, strain background (*Drosophila melanogaster*) | UASp-YFP.Rab3.T35N | Bloomington *Drosophila* Stock Center | BDSC_9766 | |
| Strain, strain background (*Drosophila melanogaster*) | UASp-YFP.Rab4.S22N | Bloomington *Drosophila* Stock Center | BDSC_9769 | |
| Strain, strain background (*Drosophila melanogaster*) | UAST-YFP.RabX4.T40N | Bloomington *Drosophila* Stock Center | BDSC_9849 | |
| Strain, strain background (*Drosophila melanogaster*) | UASp-YFP.Rab5.S43N | Bloomington *Drosophila* Stock Center | BDSC_9771 | |
| Strain, strain background (*Drosophila melanogaster*) | UAST-YFP.Rab6.T26N | Bloomington *Drosophila* Stock Center | BDSC_23249 | |
| Strain, strain background (*Drosophila melanogaster*) | UAST-YFP.RabX6.S22N | Bloomington *Drosophila* Stock Center | BDSC_9856 | |
| Strain, strain background (*Drosophila melanogaster*) | UASp-YFP.Rab7.T22N | Bloomington *Drosophila* Stock Center | BDSC_9778 | |
| Strain, strain background (*Drosophila melanogaster*) | UASp-YFP.Rab8.T22N | Bloomington *Drosophila* Stock Center | BDSC_9780 | |
| Strain, strain background (*Drosophila melanogaster*) | UASp-YFP.Rab9.S26N | Bloomington *Drosophila* Stock Center | BDSC_23642 | |
| Strain, strain background (*Drosophila melanogaster*) | UASp-YFP.Rab10.T23N | Bloomington *Drosophila* Stock Center | BDSC_9786 | |
| Strain, strain background (*Drosophila melanogaster*) | UASp-YFP.Rab11.S25N | Bloomington *Drosophila* Stock Center | BDSC_9792 | |
| Strain, strain background (*Drosophila melanogaster*) | UASp-YFP.Rab11.S25N | Bloomington *Drosophila* Stock Center | BDSC_23261 | |
| Strain, strain background (*Drosophila melanogaster*) | UAST-YFP.Rab18.S19N | Bloomington *Drosophila* Stock Center | BDSC_23238 | |
| Strain, strain background (*Drosophila melanogaster*) | UAST-YFP.Rab19.T35N | Bloomington *Drosophila* Stock Center | BDSC_9799 | |
| Strain, strain background (*Drosophila melanogaster*) | UAST-YFP.Rab21.T27N | Bloomington *Drosophila* Stock Center | BDSC_23241 | |

Appendix 1 Continued

| Reagent type (species) or resource | Designation | Source or reference | Identifiers | Additional information |
|---|---|---|---|---|
| Strain, strain background (*Drosophila melanogaster*) | UASp-YFP.Rab23.S51N | Bloomington *Drosophila* Stock Center | BDSC_9804 | |
| Strain, strain background (*Drosophila melanogaster*) | UAST-YFP.Rab26.T204N | Bloomington *Drosophila* Stock Center | BDSC_9807 | |
| Strain, strain background (*Drosophila melanogaster*) | UASp-YFP.Rab27.T25N | Bloomington *Drosophila* Stock Center | BDSC_23267 | |
| Strain, strain background (*Drosophila melanogaster*) | UASp-YFP.Rab32.T33N | Bloomington *Drosophila* Stock Center | BDSC_23281 | |
| Strain, strain background (*Drosophila melanogaster*) | UAST-YFP.Rab35.S22N | Bloomington *Drosophila* Stock Center | BDSC_9819 | |
| Strain, strain background (*Drosophila melanogaster*) | UAST-YFP.Rab39.S23N | Bloomington *Drosophila* Stock Center | BDSC_23247 | |
| Strain, strain background (*Drosophila melanogaster*) | UAST-YFP.Rab40.H25N | Bloomington *Drosophila* Stock Center | BDSC_9829 | |
| Strain, strain background (*Drosophila melanogaster*) | UAST-YFP.CG9807.T21N | Bloomington *Drosophila* Stock Center | BDSC_23257 | |
| Strain, strain background (*Drosophila melanogaster*) | UAST-YFP.CG9807.T21N | Bloomington *Drosophila* Stock Center | BDSC_23258 | |
| Strain, strain background (*Drosophila melanogaster*) | UAST-YFP.CG32673.T21N | Bloomington *Drosophila* Stock Center | BDSC_23254 | |
| Strain, strain background (*Drosophila melanogaster*) | ppk-Gal4 | Bloomington *Drosophila* Stock Center | BDSC_32078 | |
| Strain, strain background (*Drosophila melanogaster*) | ppk-Gal4 | Bloomington *Drosophila* Stock Center | BDSC_32079 | |
| Strain, strain background (*Drosophila melanogaster*) | $w^{1118}$ | Bloomington *Drosophila* Stock Center | BDSC_3605 | |
| Strain, strain background (*Drosophila melanogaster*) | TI{TI}Rab11EYFP | Bloomington *Drosophila* Stock Center | BDSC_62549 | |
| Strain, strain background (*Drosophila melanogaster*) | UASp-YFP.Rab11 | Bloomington *Drosophila* Stock Center | BDSC_50782 | |
| Strain, strain background (*Drosophila melanogaster*)ain | $hiw^{\Delta N}$ | Bloomington *Drosophila* Stock Center | BDSC_51637 | |
| Strain, strain background (*Drosophila melanogaster*) | GMR-Gal4 | Bloomington *Drosophila* Stock Center | BDSC_1104 | |
| Strain, strain background (*Drosophila melanogaster*) | *Rab11*$^{93Bi}$ | Bloomington *Drosophila* Stock Center | BDSC_4158 | |
| Strain, strain background (*Drosophila melanogaster*) | $wnd^3$/Tm3B,Sb | Bloomington *Drosophila* Stock Center | BDSC_51999 | |

Appendix 1 Continued on next page

*Appendix 1 Continued*

| Reagent type (species) or resource | Designation | Source or reference | Identifiers | Additional information |
|---|---|---|---|---|
| Strain, strain background (*Drosophila melanogaster*) | UAS-ManII-GFP | Bloomington *Drosophila* Stock Center | BDSC_65248 | |
| Strain, strain background (*Drosophila melanogaster*) | ;;UAS-ManII-TagRFP | Bloomington *Drosophila* Stock Center | BDSC_65249 | |
| Strain, strain background (*Drosophila melanogaster*) | elav-Gal4;; | Bloomington *Drosophila* Stock Center | BDSC_458 | |
| Strain, strain background (*Drosophila melanogaster*) | ;;MI{MIC}wnd[MI00494]/Tm6B | Bloomington *Drosophila* Stock Center | BDSC_39656 | |
| Strain, strain background (*Drosophila melanogaster*) | UAS-dHIP14-Tdtomato | PMID: 1803266 | Gift from Steven Stowers | |
| Strain, strain background (*Drosophila melanogaster*) | yw;;dHIP14$^{ex11}$/Tm6c | PMID: 18032660 | Gift from Steven Stowers | |
| Strain, strain background (*Drosophila melanogaster*) | Puc-LacZ | PMID: 9472024 | Gift from Bing Ye | |

