## [Editor Report · eLife assessment]

This **important** manuscript shows that axonal transport of Wnd is required for its normal degradation by the Hiw ubiquitin ligase pathway. In Hiw mutants, the Wnd protein accumulates in nerve terminals. In the absence of axonal transport, Wnd levels also rise and lead to excessive JNK signaling, disrupting neuronal function. These are interesting findings supported by **convincing** data. However, how Rab11 is involved in Golgi processing or axonal transport of Wnd is not resolved as it is clear that Rab11 is not travelling with Wnd to the axon.

---

## [Referee Report · Reviewer #1 (Public Review)]

Summary:

The manuscript by Kim et al. describes a role for axonal transport of Wnd (a dual leucine zipper kinase) for its normal degradation by the Hiw ubiquitin ligase pathway. In Hiw mutants, the Wnd protein accumulates dramatically in nerve terminals compared to the cell body of neurons. In the absence of axonal transport, Wnd levels rise and lead to excessive JNK signaling that makes neurons unhappy.

Strengths:

Using GFP-tagged Wnd transgenes and structure-function approaches, the authors show that palmitoylation of the protein at C130 plays a role in this process by promoting golgi trafficking and axonal localization of the protein. In the absence of this transport, Wnd is not degraded by Hiw. The authors also identify a role for Rab11 in the transport of Wnd, and provide some evidence that Rab11 loss-of-function neuronal degenerative phenotypes are due to excessive Wnd signaling. Overall, the paper provides convincing evidence for a preferential site of action for Wnd degradation by the Hiw pathway within axonal and/or synaptic compartments of the neuron. In the absence of Wnd transport and degradation, the JNK pathway becomes hyperactivated. As such, the manuscript provides important new insights into compartmental roles for Hiw-mediated Wnd degradation and JNK signaling control.

Weaknesses:

It is unclear if the requirement for Wnd degradation at axonal terminals is due to restricted localization of HIW there, but it seems other data in the field argues against that model. The mechanistic link between Hiw degradation and compartmentalization is unknown.

---

## [Referee Report · Reviewer #2 (Public Review)]

Summary:

Utilizing transgene expression of Wnd in sensory neurons in *Drosophila*, the authors found that Wnd is enriched in axonal terminals. This enrichment could be blocked by preventing palmitoylation or inhibiting Rab1 or Rab11 activity. Indeed, subsequent experiments showed that inhibiting Wnd can prevent toxicity by Rab11 loss of function.

Strengths:

This paper evaluates in detail Wnd location in sensory neurons, and identifies a novel genetic interaction between Rab11 and Wnd that affects Wnd cellular distribution.

Weaknesses:

The authors report low endogenous expression of wnd, and expressing mutant hiw or overexpressing wnd is necessary to see axonal terminal enrichment. It is unclear if this overexpression model (which is known to promote synaptic overgrowth) would be relevant to normal physiology.

Palmitoylation of the Wnd orthologue DLK in sensory neurons has previously been identified as important for DLK trafficking in a cell culture model.

---

## [Author Response]

The following is the authors’ response to the original reviews.

**Public Reviews:**

**Reviewer #1 (Public Review):**
Summary:The manuscript by Kim et al. describes a role for axonal transport of Wnd (a dual leucine zipper kinase) for its normal degradation by the Hiw ubiquitin ligase pathway. In Hiw mutants, the Wnd protein accumulates dramatically in nerve terminals compared to the cell body of neurons. In the absence of axonal transport, Wnd levels rise and lead to excessive JNK signaling that makes neurons unhappy.Strengths:Using GFP-tagged Wnd transgenes and structure-function approaches, the authors show that palmitoylation of the protein at C130 plays a role in this process by promoting golgi trafficking and axonal localization of the protein. In the absence of this transport, Wnd is not degraded by Hiw. The authors also identify a role for Rab11 in the transport of Wnd, and provide some evidence that Rab11 loss-of-function neuronal degenerative phenotypes are due to excessive Wnd signaling. Overall, the paper provides convincing evidence for a preferential site of action for Wnd degradation by the Hiw pathway within axonal and/or synaptic compartments of the neuron. In the absence of Wnd transport and degradation, the JNK pathway becomes hyperactivated. As such, the manuscript provides important new insights into compartmental roles for Hiw-mediated Wnd degradation and JNK signaling control.Weaknesses:It is unclear if the requirement for Wnd degradation at axonal terminals is due to restricted localization of HIW there, but it seems other data in the field argues against that model. The mechanistic link between Hiw degradation and compartmentalization is unknown.

We thank the Reviewer for valuable comments. In our revised manuscript, we have addressed reviewer ‘s comments and clarified confusions. We did not intent to imply that Rab11 directly mediates anterograde Wnd protein transport towards axon terminals. We re-worded related text throughout our manuscript to avoid confusion. Additionally, to strengthen the link between Rab11 and Wnd, we have added additional data that heterozygous mutation of wnd could rescue the eye degeneration phenotypes caused by Rab11 loss-of-function (new Figure 7C).

It is unclear if the requirement for Wnd degradation at axonal terminals is due to restricted localization of HIW there, but it seems other data in the field argues against that model. The mechanistic link between Hiw degradation and compartmentalization is unknown.

We believe that the mechanistic understanding on how Wnd protein turnover is restricted to axon/axon terminals is beyond the scope of current manuscript. We are actively investigating this interesting research question – please see our point-by-point response for details.

**Reviewer #2 (Public Review):**
Summary:Utilizing transgene expression of Wnd in sensory neurons in *Drosophila*, the authors found that Wnd is enriched in axonal terminals. This enrichment could be blocked by preventing palmitoylation or inhibiting Rab1 or Rab11 activity. Indeed, subsequent experiments showed that inhibiting Wnd can prevent toxicity by Rab11 loss of function.Strengths:This paper evaluates in detail Wnd location in sensory neurons, and identifies a novel genetic interaction between Rab11 and Wnd that affects Wnd cellular distribution.Weaknesses:The authors report low endogenous expression of wnd, and expressing mutant hiw or overexpressing wnd is necessary to see axonal terminal enrichment. It is unclear if this overexpression model (which is known to promote synaptic overgrowth) would be relevant to normal physiology.

We agree that most of our subcellular localization studies were conducted using transgenes, which may not accurately reflect endogenous protein localization. Albeit with this technical limitation, our work addresses an important mechanistic link between DLK’s axonal localization and protein turnover, in neuronal stress signaling and neurodegeneration.

Additionally, most of our experiments were done using a kinase-dead form of Wnd or with DLKi treatment (DLK kinase inhibitor). Neurons do not display synaptic overgrowth phenotypes under these experimental conditions. Thus, the changes in Wnd axonal localization are likely independent of synaptic overgrowth phenotypes.

Palmitoylation of the Wnd orthologue DLK in sensory neurons has previously been identified as important for DLK trafficking in a cell culture model.

Palmitoylation of DLK has been studied in previous works including Holland et al. 2015. These are important works. However, there are significant differences from our findings. First, inhibiting DLK palmitoylation caused cytoplasmic localization of DLK. It has been reported that expression levels of wild-type and the palmitoylation-defective DLK (DLK-CS) in axons are not different in cultured sensory neurons (Holland 2015, Figure 2A and 2B). This could be simply because DLK-CS is entirely cytoplasmic and can readily diffuse into axons – which led to the conclusion that DLK palmitoylation is essential for DLK localization on motile axonal puncta. Second, because of this cytoplasmic localization, DLK-CS failed to induce downstream signaling (Holland 2015).

However, the behavior of Wnd-CS from our study is entirely different. Wnd-CS does not show diffuse cytoplasmic localization, rather shows discrete localizations in neuronal cell bodies (Figure 2E, Figure 2-supplement 1). Furthermore, Wnd-CS is able to induce downstream signaling (Figure 4 – supplement 1 and 2). Thus, our manuscript is not an extension of previously published work. Rather, our manuscript took advantage of this unique behavior of Wnd-CS and elucidated biological function of the axonal localization of Wnd.

The authors find genetic interaction between Wnd and Rab11, but these studies are incomplete and they do not support the authors' mechanistic interpretation.

Our model describes that Wnd is constantly transported to axon terminals for protein degradation (protein turnover), and that this process is essential to keep Wnd activity at low levels to prevent unwanted neuronal stress signal. Based on this model, a failure in Wnd transport to axon terminals – as seen in Wnd-C130S or by Rab11 loss-of-function – would compromises protein degradation of Wnd, hence, results in excessive abundance of Wnd proteins. This was clearly demonstrated for Wnd-C130S (Figure 3) and for Rab11 mutants (Figure 6E), which support our model.

To strengthen the link between Rab11 and Wnd, we have added additional data in our revised manuscript, which showed that heterozygous mutation of wnd significantly rescued the eye degeneration phenotypes caused by Rab11 loss-of-function (new Figure 7C).

We did not intent to imply that Rab11 directly mediates anterograde Wnd protein transport towards axon terminals. We re-worded related text throughout our manuscript to avoid confusion.

**Recommendations for the authors:**

**Reviewer #1 (Recommendations For The Authors):**
(1) It would be interesting to overexpress Hiw in C4da neurons to see if this can degrade the C130S Wnd protein and reduce ERK signaling, or overexpress Hiw in the Rab11 mutant background to see if this can reduce the accumulation of Wnd or total Wnd levels. This could address the question of whether the reduction in Wnd turnover is due to Hiw's inaccessibility to Wnd.

Thank you for your comment. We believe this question warrants an independent line of study. Although this is beyond the scope of current work, we would like to share our findings here. We have found that overexpressing Hiw did not suppress the transgenic expression of Wnd-KD in C4da neurons regardless of cellular locations. However interestingly, the same Hiw overexpression suppressed increased Wnd-KD expression by hiw mutations in C4da neuron axon terminals. Thus, it seems that endogenous levels of Hiw in wild-type was sufficient to suppress transgenic expression of Wnd-KD, and that excessive Hiw expression does not further enhance this effect. Currently, we do not know the mechanisms underlying these observations. One possibility is that Hiw functions exclusively in the context of E3 ubiquitin ligase complex. Wu et al. (2007) found that DFsn is synaptically enriched and acts as an F-box protein of Hiw E3 ligase complex. It is possible that DFsn or some other components of Hiw E3 ligase complex determine the subcellular specificity of Hiw function. We are actively pursuing this research question currently.

(2) The authors claim that Rab11 transports Wnd to the axon terminals. However, they do not see reliable colocalization of Rab11 and Wnd at axon terminals. Can the authors see Rab11-enriched vesicles with Wnd in nerve bundles, or is the role only to sort Wnd onto a post-recycling endosome compartment that moves to axonal terminals without Rab11?

We apologize for the confusion. We did not intend to claim that Rab11 directly transports Wnd along axons. We suggested that Rab11 is necessary for axonal localization of Wnd by acting at the somatic recycling endosomes since Rab11 and Wnd extensively colocalize in the cell body but not in the axon terminals (Figure 6 and Figure 6 supplement 1). In our new “Figure 6 supplement 1”, we have now added Rab11 and Wnd colocalization in axons (segmental nerves). We also revised the text (line 294-298) “On the other hand, we did not detect any meaningful colocalization between YFP::Rab11 and Wnd-KD::mRFP in C4da axon terminals or in axons (Manders’ coefficient 0.34 ± 0.14 and 0.41 ± 0.10 respectively) (Figure 6 – supplement 1). These suggest that Rab11 is involved in Wnd protein sorting at the somatic REs rather than transporting Wnd directly.” And in Discussion (line 396-398) “These further suggest that Rab11 is not directly involved in the anterograde long-distance transport of Wnd proteins, rather is responsible for sorting Wnd into the axonal anterograde transporting vesicles.”.

(3) The authors mis-cite the Tortosa et al 2022 study which shows the exact opposite of what the authors state. Tortosa et al show DLK recruitment to vesicles through phosphorylation and palmitoylation is essential for its signaling, not the opposite, so the authors should reword that or remove the citation.

We believe the citation is correct. Tortosa et al (2022) “Stress‐induced vesicular assemblies of dual leucine zipper kinase are signaling hubs involved in kinase activation and neurodegeneration” describes that membrane association of DLK rather than palmitoylation itself is sufficient for DLK signaling activation. This is achieved by DLK palmitoylation for mammalian DLK. However, when artificially targeted to cellular membranes, palmitoylation defective DLK (mammalian DLK-CS in their study) was able to induce DLK signaling. Specifically, in their Figure 2 (K-N), when targeted to the intracellular membranes of ER and mitochondria, DLK-CS (palmitoylation defective DLK) elicited DLK signaling as shown by c-Jun phosphorylation.

**Reviewer #2 (Recommendations For The Authors):**
Major Concerns:(1) A concern is the overinterpretation of results. The authors find the accumulation of Wnd in axon terminals when they express hiw null or when they overexpress Wnd, but extrapolate that this occurs in "normal conditions" without evidence. Could the increase of Wnd in the axonal terminal be in the setting of known synaptic overgrowth associated with transgene expression?

Most of our work was conducted using a kinase-dead version of Wnd (Wnd-KD) in a wild-type background (Figure 1C and Figure 1 supplement 1). Moreover, Wnd kinase activity does not affect Wnd axonal localization in our experimental settings (Figure 1 supplement 1).

When using hiw mutant background, the larvae were treated with Wnd kinase inhibitor thus, prevented excessive axonal growth (Figure 1E, bottom right image – note that there is no axonal overgrowth in this condition). Additionally, Wnd-C130S is expressed lower levels in axon terminals than Wnd (Figure 3B) while exhibiting similar axon overgrowth (Figure 4 supplement 1B). Taken together, axonal overgrowth is unlikely affect axonal protein localization of Wnd.

(2) The interpretation of these results is based on a supposition that Rab11 anterogradely transports Wnd along axons without evidence for this. Indeed, it has been shown that Rab11 is excluded from axons in mature neurons, but can be mislocalized when overexpressed. This should be addressed in their discussion.

We apologize for the confusion. We did not intend to suggest that Rab11 directly transports Wnd along axons. We suggested that Rab11 is necessary for axonal localization of Wnd by acting at the somatic recycling endosomes since Rab11 and Wnd extensively colocalize in the cell body but not in the axon terminals (Figure 6 and Figure 6 supplement 1). In our new “Figure 6 supplement 1”, we have now added Rab11 and Wnd colocalization in axons (segmental nerves). We also revised the text (line 296-298) “On the other hand, we did not detect any meaningful colocalization between YFP::Rab11 and Wnd-KD::mRFP in C4da axon terminals or in axons (Manders’ coefficient 0.34 ± 0.14 and 0.41 ± 0.10 respectively) (Figure 6 – supplement 1). These suggest that Rab11 is involved in Wnd protein sorting at the somatic REs rather than transporting Wnd directly.” And in Discussion (line 396-398) “These further suggest that Rab11 is not directly involved in the anterograde long-distance transport of Wnd proteins, rather is responsible for sorting Wnd into the axonal anterograde transporting vesicles.”.

(3) In Figure 1, the authors should also show images of Wnd-GFSTF in wild-type (non-hiw mutations) to show endogenous Wnd levels in the axon terminal.

We have now added the figures of Wnd-GFSTF in wild-type (new Figure 1A). To show the comparable fluorescent intensities, we also re-performed hiw mutant experiment and replaced the old images.

(4) For Figure 1- Supplement, the authors state that the kinase-dead version of Wnd exhibited similar axonal enrichment in comparison to Wnd::GFP in the presence and absence of DLKi. This statement would be better supported with images specifically showing this (for example Wnd-KD::GFP compared to Wnd:GFP with DLKi and Wnd:GFP without DLKi).

We did not show the images from Wnd::GFP (DLKi) in this supplement figure because it would be redundant with Figure 1C. Rather, we presented the axonal enrichment index for Wnd::GFP (DLKi), Wnd-KD::GFP, Wnd-KD::GFP (DLKi), and Wnd-KD::GFP (DMSO) in Figure 1 supplement 1B.

Overexpressing catalytically active Wnd dramatically lowers ppk-GAL4 activity in C4da neurons thus prevents us from performing an experiment for Wnd::GFP without DLKi. In this condition, Wnd::GFP expression is barely detectable in C4da neurons.

(5) In Figure 2 - Supplement 3 the authors state that their data suggests that Wnd protein palmitoylation is catalyzed by HIP14 due to colocalization in the somatic Golgi and mutating HIP14 leads to less Wnd in the axon terminal. This statement would be better supported by evaluating Wnd's palmitoylation via immunoprecipitation in response to dHIP14 enzyme activity.

We appreciate reviewer’s comment. Although the exact identity of Wnd palmitoyltransferase might be of high interest, our study rather concerns about the biological role of Wnd axonal localization. Moreover, the identity of DLK palmitoyltransferase has been identified in mammalian cell culture and worm studies (Niu et al. 2020 “Coupled Control of Distal Axon Integrity and Somal Responses to Axonal Damage by the Palmitoyl Acyltransferase ZDHHC17”). ZDHHC17 is another name for HIP14. Our data together with these published works strongly suggest that Wnd, the Drosophila DLK might also be targeted by *Drosophila* HIP14 or dHIP14.

(6) The authors argue that palmitoylation of Wnd is essential for axonal localization of Wnd. If dHIP14 indeed palmitoylates Wnd as the authors claim, shouldn't there be a decrease in Wnd's palmitoylation within dHIP14 mutants, consequently resulting in its accumulation in the cell body rather than localization in the axonal terminal? However, Wnd is reduced at the axon terminal in dHip14 mutants, but it does not appear to increase in the cell body (Figure 2S3.C). This observation contradicts the results showing increased Wnd in the cell body presented in Figure 2. B and E. This discrepancy should be addressed.

Thank you for your comment. Our study concerns about the biological role of Wnd axonal localization. Although in an ideal model, dHIP14 mutations should prevent Wnd palmitoylation and causes subsequent cell body accumulation. However, it is highly likely that dHIP14 mutations affect a large number of protein palmitoylations – not just Wnd, which likely changes many aspect of cell functions. We envision that Wnd protein expression might be indirectly affected by these changes. In this context, mutating C130 in Wnd can be considered as more targeted approach – and our data clearly shows that such Wnd mutations render Wnd accumulation in cell bodies.

(7) Figure 3 - the authors show increased Wnd protein by western blot in WndC130S:GFP compared to Wnd::GFP. qPCR experiments to show similar mRNA expression of these two transgenes would be an important control, if it's thought that the increase of protein is due to reduction of protein degradation.

Thank you for your comment. Expressing WndC130S::GFP vs Wnd::GFP was done by GAL4-UAS system – not through endogenous wnd promoter. Thus, we do not expect different mRNA abundance of WndC130S::GFP and Wnd::GFP. However, your concern is valid for Rab11 mutants. We measured wnd mRNA abundance by RT-qPCR and found that Rab11 mutations did not increase wnd mRNA levels (Figure 6 - Supplement 2). Rather, we observed consistent reduction in wnd mRNA levels by Rab11 mutant. Please note that total Wnd protein levels were significantly increased by Rab11 mutations. We currently do not have a clear explanation. We envision that the dramatic increase in Wnd signaling (ie, JNK signal, Figure 7A) induces a negative-feedback to reduce wnd mRNA levels (line 313-317).

(8) Figure 4 Supplement - the authors report that Wnd::GFP causes robust induction of Puc-LacZ. A control without Wnd::GFP expression would be necessary to support that there was an induction.

We have added control data of UAS-Wnd-KD::GFP (new Figure 4 supplement 1A). Since this required a new side-by-side comparison of fluorescent intensities, we re-performed the full set of experiments and replaced our old data sets. The results confirmed that both Wnd::GFP and Wnd-C130S::GFP induces puc-lacZ expression.

(9) Previously it was shown that inhibiting palmitoylation of DLK prevented activation of JNK signaling (Holland et al 2015), but the authors show in Figure 4A instead an increase of JNK signaling. This discrepancy should be addressed.

The use of Wnd palmitoylation-defective mutant in our study was only possible because of different behavior of Wnd-C130S from those of palmitoylation-defective DLK. Unlike diffuse cytoplasmic localization of the palmitoylation-defective DLK in mammalian cells or in C elegans neurons, Wnd-C130S exhibited clear puncta localization in neuronal cell bodies – which extensively co-localizes with somatic Golgi complex (Figure 2E and Figure 2 supplement 1). Tortosa et al (2022) showed that palmitoylation-defective DLK (DLK-CS) can trigger DLK signaling when artificially targeted to intracellular membranous organelles (Tortosa 2022, Figure 2 (K-N)). Thus, we reasoned that unlike the palmitoylation-defective DLK from mammalian and worms, *Drosophila* DLK, Wnd might be catalytically active when mutated on Cysteine 130 because of its puncta localization.

(10) Figure 6 Supplement - the Rab11 staining is not in a pattern that would be expected with endosomes. A control of just YFP would be useful to determine if this fluorescence signal is specific to Rab11. Can endogenous Rab11 be detected in axons or in the axonal terminal?

In our model system, endogenously tagged Rab11 (TI-Rab11) does not show clear puncta patterns in segmental nerves (axons) and neuropils (axon terminals), neither colocalize with Wnd-KD. This is indeed related to the reviewer’s comment #2, which suggests that Rab11 does not form endosomes in distal axons or axon terminals in mature neurons. Expressing Rab11 transgenes exhibited some puncta structures in axons (segmental nerves) (new Figure 6 supplement 1). However, they did not show meaningful colocalize with Wnd-KD. These are consistent with our model that Rab11 acts in neuronal cell bodies for Wnd axonal transport – likely via a sorting process.

(11) There is growing evidence that palmitoylation is important for cargo sorting in the Golgi, and Rab11 is also located at the Golgi and important for trafficking from the Golgi. A mechanism that could be considered from your data is that blocking palmitoylation impairs sorting at the Golgi and trafficking from the Golgi, as opposed to impairing fast axonal transport. Indeed, Rab11 has been shown to be blocked from axons in mature neurons, making Rab11 unlikely to be responsible for the fast axonal transport of Wnd. Direct evidence of Rab11 transporting Wnd in axons would be necessary for the claim that Rab11 constantly transports DLK to terminals.

We apologize for the confusion. We did not intend to suggest that Rab11 directly transports Wnd along the axons. We suggested that Rab11 is necessary for axonal localization of Wnd by acting at the somatic recycling endosomes since Rab11 and Wnd extensively colocalize in the cell body but not in the axon terminals (Figure 6 and Figure 6 supplement 1). In our new “Figure 6 supplement 1”, we have now added Rab11 and Wnd colocalization in axons (segmental nerves). We also revised the text (line 296-298) “On the other hand, we did not detect any meaningful colocalization between YFP::Rab11 and Wnd-KD::mRFP in C4da axon terminals or in axons (Manders’ coefficient 0.34 ± 0.14 and 0.41 ± 0.10 respectively) (Figure 6 – supplement 1). These suggest that Rab11 is involved in Wnd protein sorting at the somatic REs rather than transporting Wnd directly.” And in Discussion (line 394-398) “These further suggest that Rab11 is not directly involved in the anterograde long-distance transport of Wnd proteins, rather is responsible for sorting Wnd into the axonal anterograde transporting vesicles.”.